# AN OLD DOG CAN LEARN (SOME) NEW TRICKS: A TALE OF A THREE-DECADE OLD ARCHITECTURE

## ABSTRACT

Designing novel architectures often involves combining or extending familiar components such as convolutions and attention modules. However, this approach can obscure the fundamental design principles as the focus is usually on the entire architecture. Instead, this paper takes an unconventional approach, attempting to rejuvenate an old architecture with modern tools and techniques. Our primary objective is to explore whether a 30-year-old architecture can compete with contemporary models, when equipped with modern tools. Through experiments spanning image recognition datasets, we aim to understand what aspects of the architecture contribute to its performance. We find that while an ensemble of ingredients bears significance in achieving commendable performance, only a few pivotal components have a large impact. We contend that our discoveries offer valuable insights for creating cutting-edge architectures.

## 1 INTRODUCTION

Deep neural networks (DNNs) have achieved remarkable breakthroughs in image recognition (He et al., 2016; Huang et al., 2017) and beyond (Vaswani et al., 2017; Radford et al., 2022). Architecture design lies at the forefront of this success with a plethora of novel architectures proposed every single year. However, it is unclear what benefits newer architectures bring when compared to their older counterparts. Our goal in this paper is precisely to focus on this topic by posing the following question:

*How can a 30-year-old architecture fare when compared to modern architectures?*

We confirm experimentally that a 30-year-old architecture (Shin & Ghosh, 1991) (which has been sidelined for so long) cannot compete with modern models on its own. There are two key paths we can take: (a) design novel techniques to make this competitive, (b) use off-the-shelf techniques to improve its performance. Designing novel techniques is the dominant method in the literature, such as in the influential works of Highway networks (Srivastava et al., 2015) or MLP-Mixer (Tolstikhin et al., 2021). However, we argue that there is an abundance of techniques available the last few years and yet few insights into how each technique fares in other architectures or its role in generalization in new datasets. Therefore, in this work, we focus on using *off-the-shelf techniques*. Since those techniques are designed for other architectures, this enables us to obtain fresh insights the techniques that are critical in architecture design.

### 1.1 CONTRIBUTIONS AND FINDINGS

Our work aims to uncover key insights on the critical techniques contributing to the exceptional performance of DNNs. Our analysis uses the core block of Pi-Sigma (Shin & Ghosh, 1991) as the starting point. Notably, the design of Pi-Sigma deviates significantly from the prevailing paradigms commonly employed in image recognition, rendering it an excellent testbed for our investigation. We explore the potential for further enhancements of the block by incorporating well-established techniques such as normalization schemes and modern data augmentation approaches during the training process. Our aim is to evaluate the efficacy of these techniques in improving the performance of Pi-Sigma.

Concretely, we showcase how the original model can be augmented to match the performance of standard baselines, such as ResNet (He et al., 2016), on ImageNet (Deng et al., 2009). To achieve that, we use well-established techniques from the literature. Our investigation of those techniques reveals several insights:

- Only a few techniques, namely the a) training algorithm, b) normalization scheme, and c) data augmentation with label noise, play a critical role consistently across benchmarks.

- However, data augmentation should be combined with a deeper architecture, so we consider depth as enabler of an improved performance.

- Skip connections seem to offer advantages in terms of optimization, but their role in the generalization should be further scrutinized.

- There are non-trivial correlations when adding techniques, which means that the ordering that techniques are added directly impacts the performance.

Through meticulous validation on *eighteen diverse recognition datasets* encompassing natural-scene and medical images, our refined Pi-Sigma model, called RPS, demonstrates exceptional performance that closes the gap with well-established baselines, including the highly regarded ResNet. These compelling findings deepen our understanding of the underlying inductive bias embedded within the architecture and hold promise for driving further advancements in both empirical and theoretical research.

We intend to make the source code publicly available upon the acceptance of the paper to enable further studies on understanding the inductive bias of models.

## 1.2 RELATED WORK

The field of neural architecture design has been thriving for several decades, with numerous seminal works. One of the earliest and most influential works is that of Rosenblatt, who introduced the concept of the perceptron (Rosenblatt, 1958). This model laid the foundation for the development of learning algorithms and demonstrated the potential of neural networks in solving classification problems. Among the early architectures, the multi-layer perceptron (MLP) and the radial basis function network (RBFN) (Park & Sandberg, 1991) are notable examples. The MLP is a feedforward neural network composed of multiple layers of nodes, each of which performs a weighted sum of the inputs followed by a nonlinear activation function. The RBFN, on the other hand, employs radial basis functions to model the input-output relationship.

The resurgence of neural networks has led to an explosion of new blocks (Srivastava et al., 2015; Szegedy et al., 2015; Larsson et al., 2017; Li et al., 2017; Chrysos et al., 2020; Bolya et al., 2022). The GoogLeNet (Szegedy et al., 2015) introduced the concept of the inception module, which consists of multiple convolutional filters of different sizes, concatenated in parallel to capture multi-scale features. Another influential architecture is ResNet (He et al., 2016), which introduced residual connections and enabled the training of much deeper neural networks. ResNet has since become a cornerstone of deep learning research and has inspired numerous follow-up works, such as the recent Hornet architecture (Rao et al., 2022), which extends the residual connection concept with a dynamic routing mechanism.

DenseNet (Huang et al., 2017), which creates dense connections between different layers, allows for the direct flow of information between all layers and promotes feature reuse, resulting in improved accuracy and reduced parameter count. The SqueezeNet block (Iandola et al., 2016) is also designed for efficient use of parameters, with a significant reduction in model size without sacrificing accuracy. The Squeeze-and-Excitation module improves the representational power of neural networks by adaptively modifying the feature maps (Hu et al., 2018).

Our work differs from the aforementioned studies that mainly introduce a new block or seek to establish scaling laws for existing architectures (Simonyan & Zisserman, 2015; Tan & Le, 2019; Radosavovic et al., 2020). Instead, our study does not aim to propose a novel architecture. In terms of motivation, the works of Ding et al. (2021); Liu et al. (2022) are more aligned with ours, as they re-evaluate the convolutional networks and aim to "modernize" them using well established techniques. However, our objective is to delve deeper into architecture design and explore how previously used techniques and tools, such as the skip connection or normalization layers (Ba et al., 2016) can be adapted to enhance old architectures.

Our work is connected to the extensive body of research on explicit and implicit regularization techniques employed in the training of DNNs. These techniques play a pivotal role in enhancing the generalization of DNN models in challenging benchmarks. One prominent category of regular-

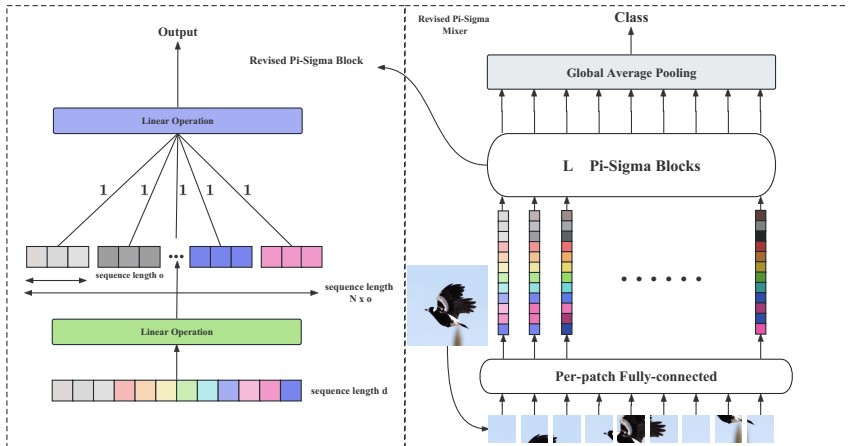

Figure 1: (Left) The original Pi-Sigma visualizing Eq. ($\Pi-\Sigma$). Given a vector as input, it uses two linear operations. After the first linear operation, the resulting vectors are multiplied together and another linear operation is applied. (Right) Abstract schematic of processing an image. Concretely, given an input image, the network splits the image into non-overlapping patches, which are then fed into $L$ revised Pi-Sigma blocks.

ization methods is data augmentation (Shorten & Khoshgoftaar, 2019; Lin et al., 2019; Xie et al., 2020). Notable advancements in data augmentation techniques include Mixup (Zhang et al., 2018), AutoAugment (Cubuk et al., 2019), and CutMix (Yun et al., 2019). In addition to data augmentation, various feature normalization schemes have emerged as influential regularization techniques, such as dropout (Srivastava et al., 2014) or dropblock (Ghiasi et al., 2018). In this work, we aim to use standard techniques as plug-in methods and not introduce a specific regularization scheme.

## 2 REVISED PI-SIGMA

We succinctly describe the notation and the original Pi-Sigma block and then develop a step-by-step improvement of the Pi-Sigma.

**Notation**: Vectors (or matrices) are indicated with lowercase boldface letters e.g., $\boldsymbol{w}$ (or $\boldsymbol{W}$). Tensors are denoted with calligraphic boldface letters, e.g., $\boldsymbol{\mathcal{W}}$. Given a matrix $\boldsymbol{W}$, $w_{i,j}$ denotes the scalar element in the $i^{\text{th}}$ row and the $j^{\text{th}}$ column. We assume each training and testing sample is an image $\boldsymbol{\mathcal{X}} \in \mathbb{R}^{d_1 \times d_2 \times 3}$. We partition the image into non-overlapping patches (similarly to transformers (Dosovitskiy et al., 2021) or MLP-Mixer (Tolstikhin et al., 2021)). Each patch is vectorized and expressed as $\boldsymbol{x} \in \mathbb{R}^d$, while the whole image can be expressed as a sequence of those $m$ patches, i.e., $\boldsymbol{X} \in \mathbb{R}^{m \times d}$. Unless explicitly mentioned otherwise, our analysis below assumes the input is a vectorized patch $\boldsymbol{x}$.

**Original Pi-Sigma**: Let us firstly describe the original Pi-Sigma (Shin & Ghosh, 1991). The Pi-Sigma aims to capture high-order correlations without increasing the parameters exponentially as a full polynomial expansion would (Chrysos et al., 2019). To achieve that, a Pi-Sigma block utilizes the product of intermediate representations in a single layer. Mathematically, the block is expressed as:

$$h_{i,j} = \sum_{k=1}^{d} w_{i,j,k} x_k \ , \qquad y_i = \sigma \left( \prod_{j=1}^{N} h_{i,j} \right), \qquad (\Pi-\Sigma)$$

where $\sigma$ denotes an elementwise activation function (typically sigmoid), $\boldsymbol{\mathcal{W}}$ is a learnable parameter and $\boldsymbol{y} \in \mathbb{R}^o$ is the output of the block. By changing the value of $N \in \mathbb{N}$, we control the order of the correlations captured. As such, a single block of Eq. ($\Pi-\Sigma$) is used for approximating functions.

Below, we will discuss the techniques employed to enhance the performance of the Pi-Sigma network, drawing inspiration from neural architecture design principles. We utilize Eq. ($\Pi-\Sigma$) as a fundamental building block and stack multiple blocks sequentially. However, merely stacking these blocks alone leads to unsatisfactory training outcomes. To overcome this challenge, we incorporate

well-established techniques such as skip connections and normalization schemes to improve the performance. The application of these techniques results in a significant enhancement in the overall empirical performance of the Pi-Sigma network, as demonstrated through our rigorous experimental validation. In the following paragraphs, we provide a detailed explanation of each technique that is added progressively.

**Technique 1** *The first two improvements on Eq. ($\Pi$–$\Sigma$) will be a) connecting sequentially L Pi-Sigma blocks, and b) adding skip connections. Then, the $l^{th}$ block with $l \in [1, L]$ becomes:*

$$h_{i,j}^{(l)} = \sum_k w_{i,j,k}^{(l)} y_i^{(l-1)} \,, \qquad y_i^{(l)} = \sigma \left( \prod_j h_{i,j}^{(l)} \right) + \sum_k v_{i,k}^{(l)} y_i^{(l-1)}, \qquad (1)$$

*where $\boldsymbol{y}^{(0)} = \boldsymbol{x}$ is the input and $\boldsymbol{y} = \boldsymbol{y}^{(L)}$ is the output. The learnable parameter $\boldsymbol{V}$ scales the previous output $y_i^{(l-1)}$ to match the dimensions of the output in the block $l$.*

**Technique 2** *A learnable parameter $\alpha^{(l)} \in \mathbb{R}$ is utilized to scale the product of $\boldsymbol{h}^{(l)}$ terms.*

**Technique 3** *A normalization function, denoted as $\rho$, is employed on the product of $\boldsymbol{h}^{(l)}$ terms.*

**Technique 4** *A learnable square matrix $\boldsymbol{S}^{(l)} \in \mathbb{R}^{o \times o}$ is added after the normalization function and the formulation of block $l$ becomes:*

$$h_{i,j}^{(l)} = \sum_k w_{i,j,k}^{(l)} y_i^{(l-1)} \,, \qquad y_i^{(l)} = \alpha^{(l)} \cdot \boldsymbol{S}^{(l)} \cdot \rho \left( \sigma \left( \prod_j h_{i,j}^{(l)} \right) \right) + \sum_k v_{i,k}^{(l)} y_i^{(l-1)} \,. \quad (2)$$

The aforementioned techniques capture correlations inside a single patch $\boldsymbol{x}$. However, typically a single patch carries only partial information about the final prediction. We use below two techniques to augment the information captured from a single vectorized patch: a) we use a convolutional layer in the input space to capture spatial information inside the patch (cf. Technique 5), b) we express correlations across patches, which can be thought of as capturing spatial information across the image (cf. Technique 6).

**Technique 5** *A learnable convolutional filter is used in the original input image $\boldsymbol{\mathcal{X}}$.*

**Technique 6** *In addition to the previous blocks, we create another block with a similar structure. Let us consider the transpose input matrix $\boldsymbol{X}^T \in \mathbb{R}^{d \times m}$ as a sequence of d elements $\hat{\boldsymbol{x}} \in \mathbb{R}^m$. Each $\hat{\boldsymbol{x}}$ captures information across patches in a uniform way. For instance, the first row of $\boldsymbol{X}^T$, which is the first $\hat{\boldsymbol{x}}$, corresponds to the top-left pixel of every patch. Next, we treat each $\hat{\boldsymbol{x}}$ as input to a Pi-Sigma block. All of the aforementioned techniques are still applicable to this "transpose" block.*

**Remarks**: The overall schematic of the model is depicted in Fig. 1, while the following remarks apply for the aforementioned techniques:

- All the techniques have been used previously in the literature, sometimes with a different goal than ours. Namely, Technique 2 has previously appeared as a way to avoid normalization layers (Brock et al., 2021). However, we find that $\alpha$ improves the performance in our case.
- The transpose block in Technique 6 has been previously utilized by the MLP-Mixer (Tolstikhin et al., 2021). Even if the format of the block differs, the idea is similar, i.e., capturing correlation across patches.
- The intermediate representations $\boldsymbol{y}^{(l)}$ can differ in dimension, however, to simplify the notation we assumed above that they are all in the $\mathbb{R}^o$. As such, we will refer to $o$ as the hidden dimension in the experimental part.

**Learning the revised model**: We employ standard data augmentation techniques and learning algorithms, which are detailed below in the training setup and in Appendix A. All of those augmentation techniques are standard in the recognition literature. The final model, called RPS, is learned using standard gradient-based learning algorithms, similar to the rest baselines.

## 3 EXPERIMENTS

In this section, we present experimental results to validate the performance of Pi-Sigma. Firstly, we describe the experimental setup, followed by experimental validation of the techniques mentioned on Sec. 2. Sequentially, we conduct an ablation study on important hyperparameters. Furthermore, we compare our revised Pi-Sigma, referred to as RPS, with a range of neural networks on ImageNet in Sec. 3.4 and medical imaging in Sec. 3.6. We also exhibit the benefits of pre-training in Sec. 3.5. Due to space limitations, further information on the benchmarks and additional experiments can be found in the Appendices A and B respectively. Even though we augmented the architecture in order to improve the clean accuracy, we also scrutinize the improvement in robustness to diverse corruptions in Appendix D.

### 3.1 EXPERIMENTAL SETUP

Unless otherwise mentioned explicitly, the final model, called RPS, consists of 75 blocks of Technique 6, while we set $N = 3$, patch size 4 and $\rho$ as the Layer Normalization (Ba et al., 2016). The patch size and the width differ per experiment and are described in detail in the Appendix A.

**Learning**: We train our model using AdamW (Loshchilov & Hutter, 2019) with batch size 128. We use a linear warmup and cosine decay, while the initial learning rate is 3e-4 and gradually drops to 1e-5 in 300 epochs. We also use label smoothing (Szegedy et al., 2016) with probability 0.15 and commonly used data-augmentation strategies such as auto-augment (Cubuk et al., 2019) and random erasing with probability 0.2. These settings are commonly used in methods (Tolstikhin et al., 2021). On ImageNet, we train our model using eight NVIDIA A100 GPUs. The batch size is set to 2,048. The initial learning rate is set to 1e-3. The rest settings remain the same.

### 3.2 FROM PI-SIGMA TO RPS

We present a comprehensive validation study on the widely recognized Cifar-10 and Cifar-100 datasets, showcasing the potential for significant improvements in the original Pi-Sigma model presented in Eq. ($\Pi-\Sigma$). Our experimental findings, as illustrated[1] in Fig. 2, elucidate the incremental contributions of each technique employed, resulting in a notable enhancement in the accuracy of the original model. The techniques are applied in the prescribed order outlined in Sec. 2, with alternative orderings also yielding similar outcomes, as demonstrated in the Appendix. The first notable improvement arises in the second row, where we replace the SGD optimizer with AdamW, incorporating gradient clipping and a cosine scheduler. Furthermore, we observe considerable gains from increasing the network depth (along with skip connections) and introducing a scalar parameter $\alpha^{(l)}$ per layer. A crucial enhancement arises from adding layer normalization layers, which help to stabilize the training process. Although the use of a convolutional layer in the input space has a marginal effect, incorporating data augmentations leads to a significant improvement in the performance, as observed in the performance in the penultimate and the last rows in Fig. 2. These findings collectively emphasize the potential of these techniques to enhance the original Pi-Sigma model, shedding light on effective strategies for improving its performance.

We conduct an additional experiment where we apply progressively the top-3 performing techniques from Fig. 2. The goal is to evaluate whether those three ideas can improve the performance without the rest techniques. The results are reported in Table 1. The second column corresponds to the original model trained with SGD and a fixed learning rate, while subsequent columns feature improvements due to replacing SGD with AdamW and adding a learning rate scheduler. Moreover, the fourth column introduces the Layer Normalization and the fifth column integrates additional data augmentations. Interestingly, significant gains are observed for the first two modifications, whereas the third modification with the data augmentations has a negative effect. This behavior can be explained by the limited depth of the single block architecture. Our findings suggest that even though deeper networks may benefit from increased regularization via data augmentations, shallow architectures are less reliant on data augmentations.

---

[1] The tables with the complementary results are reported in Appendix B.1 due to limited space in the paper.

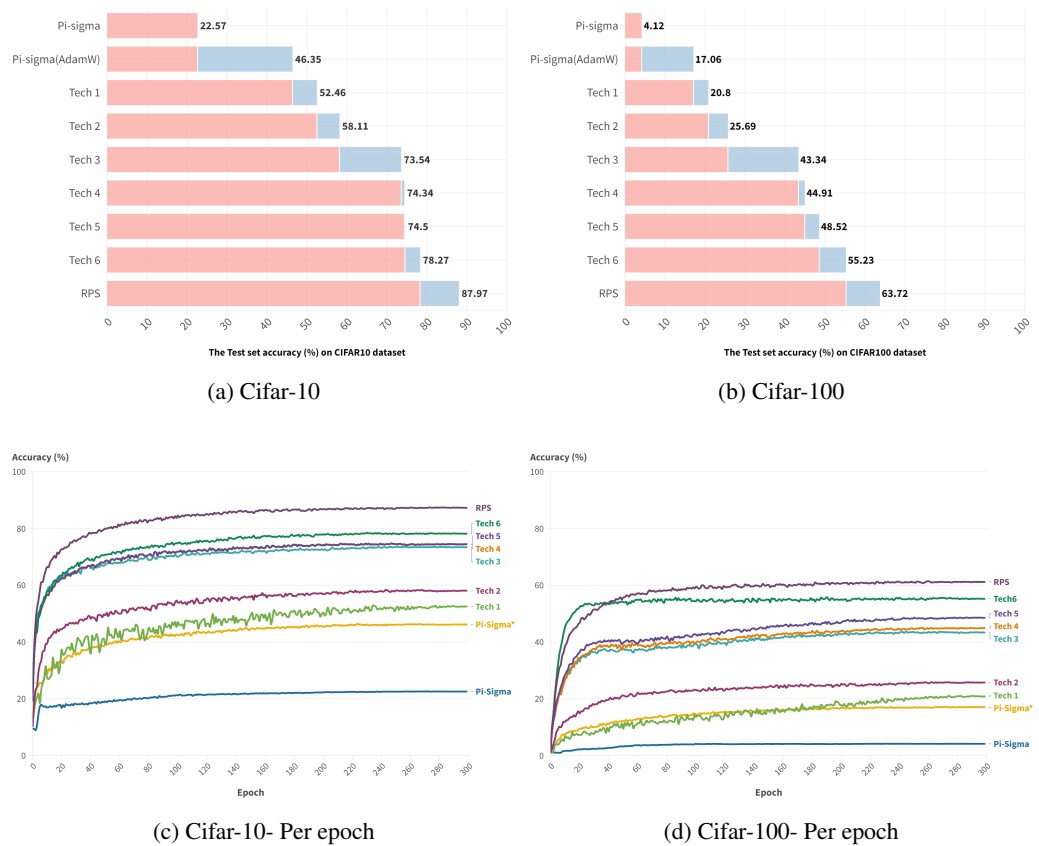

(a) Cifar-10

(b) Cifar-100

(c) Cifar-10- Per epoch

(d) Cifar-100- Per epoch

Figure 2: (a), (b) Step-by-step improvement of Pi-Sigma on Cifar-10 and Cifar-100. The blue color depicts the absolute improvement in accuracy when adding techniques incrementally. RPS is the final model, i.e., Technique 6, plus data augmentation and Label smoothing. Notice that each additional technique has a positive contribution to the final performance. We observe that the pivotal factors responsible for substantial performance gains are the learning algorithm (highlighted in the second rows) and the application of Technique 3 in conjunction with the employment of data augmentation (depicted in the last row). (c), (d) Accuracy per epoch. Notice that many patterns emerge similarly in the two datasets as techniques are added incrementally, while all of the variants improve (even marginally) until the 300th epoch. Interestingly, Technique 2 smooths the testing accuracy from the (minor) oscillation that it had previously.

Table 1: Performance evaluation with Pi-Sigma variants. We progressively incorporate the top-performing strategies from the prior analysis of Fig. 2. The second column uses the original Pi-Sigma using SGD with a fixed learning rate. The rest columns use the original model learned with AdamW and with a scheduler. The fourth column adds the Layer Normalization and the fifth adds the data augmentation. Notice that the performance in the first two (i.e., learning algorithm and normalization scheme) vastly improves. On the contrary, the data augmentation does not have the same effect, which is attributed to the fact that the model includes a single block.

| Dataset | SGD† | AdamW + scheduler | LN ⋆ | Data augmentation ⋆ |
|---|---|---|---|---|
| Cifar-10 | 0.244 | 0.464 | 0.569 | 0.557 |
| Cifar-100 | 0.041 | 0.171 | 0.284 | 0.265 |

The '†' denotes the variant that is run with vanilla SGD with a fixed learning rate, while the rest columns use AdamW + scheduler.

The symbol '⋆' denotes the setting from the previous column plus the new technique.

Table 2: Ablation study on the number of the Pi-Sigma blocks. Beyond the first few blocks, the rest blocks have diminishing returns in terms of performance.

Table 3: Ablation study on the width of a layer. Different widths share a similar performance.

| Depth | 3 | 9 | 18 | 36 | 54 | 72 | 75 | 90 |
|---|---|---|---|---|---|---|---|---|
| Acc. | 0.748 | 0.855 | 0.868 | 0.871 | 0.871 | 0.877 | **0.893** | 0.881 |

| Width | 128 | 192 | 256 | 384 |
|---|---|---|---|---|
| Acc. | 0.868 | **0.893** | 0.882 | 0.891 |

## 3.3 ABLATION STUDIES

We conduct three self-evaluation experiments on Cifar-10 to assess the effect of critical hyperparameters. The results of Fig. 2 indicate that depth is an important parameter, along with the normalization function. In addition, we evaluate the effect of the width of a layer.

**Normalization scheme**: We experiment with two influential normalization schemes, i.e., Layer Normalization and Batch Normalization (Ioffe & Szegedy, 2015). The accuracy in the case of Batch Normalization drops to $0.870$, while with Layer Normalization we obtain an accuracy of $0.893$. We believe that more elaborate normalization schemes could further improve the performance, but the goal of this work is to use standard tools from the literature.

**Depth**: We assess the impact of the number of Pi-Sigma blocks. A block is composed of Eq. ($\Pi - \Sigma$) on (a) the vectorized image patch and (b) the 'transpose' (i.e., Technique 6). In Table 2, we exhibit how the depth impacts the performance. Notice that there is a considerable improvement moving from 3 to 9 blocks, while the improvements from that point forward are incremental. Our experiment demonstrates that a depth of $75$ yields the best performance, so we use this in the rest of the paper.

**Width**: We experiments with varying width size for the layer of Eq. ($\Pi - \Sigma$). The results in Table 3 exhibit that RPS is resilient to the change of the width. In the rest of the paper, we use a width size of $192$, since this performs slightly better.

## 3.4 COMPARISON WITH DIVERSE NETWORKS ON IMAGENET

We conduct experiments with diverse networks on ImageNet (Deng et al., 2009), a major benchmark in image recognition. Five methods are used as baselines. We utilize ResNet-18 as the default baseline in recognition. Similarly to the Pi-Sigma that captures high-order correlations, $\Pi$-Nets (Chrysos et al., 2020) are utilized as a baseline, since they also can capture high-order correlations. We also add indicative architectures used for recognition, such as SimplePatch (Thiry et al., 2021) and MLP-Mixer (Tolstikhin et al., 2021) .For each baseline, we report the accuracy reported in the original paper. The results in Table 4 illustrate that the revised model crosses the $70\%$ threshold of ResNet-18. Note that this performance is achieved with a 23-million-parameter model. Importantly, scaling RPS further seems to provide a further boost, reaching a competitive performance of $76.87\%$. This confirms the critical role of depth in challenging benchmarks.

## 3.5 PRE-TRAINING ON IMAGENET-21K

A reasonable question is whether the accuracy of RPS improves further when we use a larger dataset for pre-training. To assess the hypothesis, we conduct pre-training on ImageNet-21K, a larger dataset of 21,841 classes (a superset of the 1,000 ImageNet classes) with around 14M images and then fine-tune the pre-trained model on ImageNet for evaluation. More specifically, we pre-train the model on ImageNet-21K for 300 epochs using AdamW with a batch size of $1,024$. We use a linear warmup and cosine decay, while the initial learning rate is 1e-3. Then, we fine-tune ImageNet-21K pre-trained models on ImageNet for another 100 epochs. We use AdamW, a cosine learning rate schedule, a learning rate of 5e-5, no weight decay, and a batch size of 512. The default pre-training, fine-tuning, and testing resolution is $224 \times 224$. Common data-augmentation strategies and label smoothing are used during pre-training and fine-tuning following Tolstikhin et al. (2021).

As we can see from Table 5, pre-training on larger datasets significantly improves the performance of the proposed RPS from $71.45\%$ to $74.96\%$. Similar improvement can be also observed when the patch size is decreased from $16 \times 16$ to $14 \times 14$. Notice that the large RPS from Table 4 has a comparable accuracy to the last model of Table 5, while the latter one has almost half of the parameters of the former. In other words, we find that larger pre-training is a beneficial technique and the results reported with pre-training should be clearly identified as such.

Table 4: Experimental results on ImageNet. The results indicate that the proposed RPS can reach the 70% threshold, which is the baseline performance of the well-established ResNet-18, using more parameters though (23 million parameters). Importantly, scaling RPS further improves the performance, performing favorably to MLP-Mixer and ResNet-50. The symbol $o$ in our model signifies the hidden size in the block, while $L$ denotes the number of blocks.

| Model | # Training Epochs | Resolution | ImageNet Top-1 Accuracy | FLOPs (G) | # Par (M) |
|---|---|---|---|---|---|
| ResNet-18 | 100 | 224 | 69.90 | 1.82 | 11.7 |
| ResNet-50 | 100 | 224 | 76.55 | 4.12 | 25.6 |
| Π-Nets | 100 | 224 | 65.20 | 1.90 | 12.3 |
| ViT-B/16 | 300 | 224 | 77.91 | 17.60 | 86.6 |
| SimplePatch | 60 | 64 | 39.00 | - | 85.5 |
| MLP-Mixer-B/16 | 300 | 224 | 76.44 | 11.60 | 59.0 |
| Pi-Sigma (single block) | 300 | 224 | 7.62 | 0.09 | 0.6 |
| RPS ($o = 128$, $L = 51$) Small | 300 | 224 | 64.31 | 1.73 | 11.6 |
| RPS ($o = 192$, $L = 75$) Medium | 300 | 224 | 71.45 | 4.48 | 23.3 |
| RPS ($o = 256$, $L = 141$) Large | 300 | 224 | 76.87 | 12.94 | 59.7 |

Coloring denotes the category of the method: ▮ CNN-based ▮ MLP-based

Table 5: Additional experimental results on ImageNet with ImageNet-21K pre-training. The results indicate that when the patch size decreases from $16 \times 16$ to $14 \times 14$, the performance of the proposed RPS increases from $71.45\%$ to $73.26\%$. Importantly, pre-training on ImageNet-21k further improves the performance. The symbol $o$ in our model signifies the hidden size in the block, while $L$ denotes the number of blocks.

| Model | Pre-train | # Training Epochs | Resolution | ImageNet Top-1 Accuracy | FLOPs (G) | # Par (M) |
|---|---|---|---|---|---|---|
| RPS ($o = 192$, $L = 75$, patch size = 16) | From scratch | 300 | 224 | 71.45 | 4.48 | 23.3 |
| RPS ($o = 192$, $L = 75$, patch size = 16) | ImageNet-21k | 100 | 224 | 74.96 | 4.48 | 23.3 |
| RPS ($o = 192$, $L = 75$, patch size = 14) | From scratch | 300 | 224 | 73.26 | 6.70 | 31.3 |
| RPS ($o = 192$, $L = 75$, patch size = 14) | ImageNet-21k | 100 | 224 | 76.08 | 6.70 | 31.3 |

## 3.6 EXPERIMENTAL VALIDATION ON MEDICAL IMAGES

We aim to extend the application of RPS outside of the image recognition of natural scenes to the critical domain of medical imaging. We benchmark the performance of RPS against the popular baseline of ResNet18 in the MedMNIST challenge (Yang et al., 2021). This challenge includes eleven different datasets with diverse types of images (e.g., X-rays, MRI scans) and from different parts of the body (e.g., tissues, retina, skin cells). Each dataset contains a different number of classes. We follow the protocols of the challenge for executing all experiments: each model is trained for 100 epochs without any data augmentation. In this experiment, we utilize a patch size of 2 and a hidden size of 128 for a total of 13 million parameters. The results[1] in Fig. 3a demonstrate how RPS outperforms in all cases Pi-Sigma. RPS also performs favorably to ResNet in six out of the eleven datasets. Along with the results in Appendix B.10, we confirm that RPS outperforms the original model, but is also competitive to ResNet in certain cases.

## 3.7 IMAGE RECOGNITION WITH LIMITED TRAINING DATA

Recent methods rely on vast amounts of data in the scale of ImageNet to perform well. To scrutinize RPS, we assess its performance in the presence of limited training data. We progressively reduce the number of samples per class from 5000 to 100, when trained on Cifar-10. The rest of the details remain similar to Appendix B.10. The performance[1] in Fig. 3b confirms that RPS outperforms Pi-Sigma even in the presence of limited data.

## 3.8 DISCUSSION

Our empirical evaluation exhibits that RPS performs on par with recent architectures on various benchmarks, vastly improving the performance of the dated Pi-Sigma. To achieve that performance,

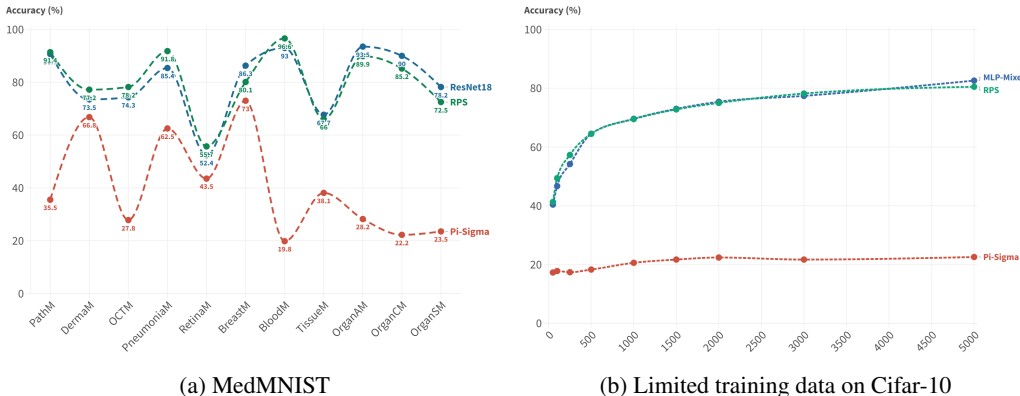

(a) MedMNIST          (b) Limited training data on Cifar-10

Figure 3: (a) Performance in MedMNIST challenge. Each element in the x-axis denotes a separate dataset from the challenge. Overall, there is a substantial improvement from Pi-Sigma to RPS. (b) Performance in Cifar-10 when trained with limited data (as denoted in the x-axis). Importantly, RPS outperforms Pi-Sigma, while it performs on par with MLP-Mixer.

we quantify the improvement of each technique in Sec. 3.2 for Cifar-10/Cifar-100 and Appendix B.3 for STL-10. The most important insight is the techniques that are critical: a) the training algorithm, b) the normalization scheme, c) the data augmentation, d) the depth. Additional techniques, such as the pre-training can have a large impact, especially on more diverse datasets. We believe our insights can lead to a deeper understanding of the inductive bias of each technique.

Beyond the critical techniques, our study reveals a number of surprising findings. Firstly, we observe that the skip connections might not contribute in the performance improvement, a phenomenon that warrants further exploration. This is contrary to the existing understanding in theoretical works (Tirer et al., 2022). We believe this can offer fresh insights for a refined theoretical analysis on skip connections. Furthermore, we observe that existing initializations tend to perform similarly on RPS as indicated in Appendix B.2. Similarly, in Appendix G, we highlight other well-established techniques that did not provide any improvement. Lastly, we observe that the contribution of each technique depend considerably on the data distribution and the already added techniques. The transpose block (i.e. Technique 6) is a prime example. Fig. 2(a) exhibits a relative improvement of $5\%$ on Cifar-10 when the transpose block is added, while in Fig. 2(b) the relative improvement is $14\%$ on Cifar-100. The issue is further exacerbated by the order of which components are tried out, e.g., notice how in Table 13 the improvement from the transpose block is large even on Cifar-10 by assuming an alternative ordering of components. We believe this is a key point indicating *how hard the design of an architecture is*.

## 4    CONCLUSION

In this work, we conduct a comprehensive analysis of Pi-Sigma, a neural architecture with a rich history spanning three decades, and assess its performance within modern benchmark settings. Through the strategic integration of established techniques such as layer normalization and skip connections, we demonstrate a notable enhancement in the capabilities of Pi-Sigma. Moreover, we meticulously examine the influence of contemporary learning methods, including data augmentation and recent optimizers. Intriguingly, our findings reveal that the refined architecture achieves performance levels comparable to recent architectures. Our study facilitates a comprehensive understanding of the benefits derived from a holistic evaluation of models, shedding light on the key techniques that drive architectural advancements. We anticipate that our investigation will contribute to the field, empowering researchers to make informed decisions when devising cutting-edge architectures for diverse applications.

**Limitations**: Our analysis is empirical, while a thorough theoretical understanding of the models might indicate additional differences from the baseline models. In addition, our evaluation relies on well-known benchmarks in image recognition and we cannot guarantee the validity of the Pi-Sigma beyond recognition. We encourage the community to further study those models for different tasks.

## REPRODUCIBILITY STATEMENT

We intend to release the open source code of our model upon the acceptance of our work. We use only publicly available benchmarks in this work to ensure that a practitioner can reproduce our studies. We also describe the hyperparameters used in this work and try to describe all the techniques we use in details along with supporting ablation studies.

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

Hongyi Zhang, Moustapha Cisse, Yann N Dauphin, and David Lopez-Paz. mixup: Beyond empirical risk minimization. 2018. 3

CONTENTS OF THE APPENDIX

A number of additional experimental details and results are added to the appendix for elaborating on the performance of Pi-Sigma. The exact details in the Appendix are as follows:

- Appendix A is devoted to the experimental setup details, including the datasets and benchmarks used.
- Additional experimental results are exhibited in Appendix B.
- Appendix C explores the impact of different techniques on the confidence of the classifier.
- Appendix D explores the robustness of the revised model to diverse corruptions.
- Appendix E extends the evaluation of the different techniques beyond the Pi-Sigma architecture.
- Appendix F refers to the broader impact and ethics statement of our work.
- Some less successful techniques are reported in Appendix G.

## A  EXPERIMENTAL DETAILS

We refer below to details on the datasets and the comparison methods.

**Datasets**: The following datasets are utilized as benchmarks in this work:

- *Cifar-10* (Krizhevsky et al., 2014) comprises of $60,000$ images depicting objects in natural scenes, each with a resolution of $32 \times 32 \times 3$ pixels and categorized into one of $10$ classes. Due to its suitability as a benchmark for image recognition, Cifar-10 has been widely adopted for research purposes.
- *Cifar-100* (Krizhevsky et al.) contains $100$ object classes with $600$ ($500$ for training, $100$ for testing) images annotated per class. Each image has a resolution of $32 \times 32 \times 3$ pixels.
- *STL-10* (Coates et al., 2011) contains $10$ object classes that share similarities with Cifar-10. However, each image has a higher resolution, and there are less images in total than Cifar-10. In particular, each image is of resolution $96 \times 96$. Each class includes $500$ images for training and $800$ images for testing.
- The Street View House Numbers dataset, known as *SVHN* (Netzer et al., 2011), comprises of $100,000$ digit images, of which $73,257$ are used for training. The images depict color house numbers and are classified into $10$ classes, each corresponding to a digit between $0$ and $9$. SVHN images exhibit diverse backgrounds and scales.
- *Tiny ImageNet* (Le & Yang, 2015) contains $200$ object classes, where each image is of resolution $64 \times 64$. Each class includes $500$ annotated images.
- Oxford flowers (abbreviated as *O-Flowers* here) (Nilsback & Zisserman, 2008) contains $102$ categories of flowers annotated, with $10$ training images per class and $10$ validation images per class.
- *ImageNet* (Deng et al., 2009) contains $1,000$ object classes. The dataset includes over one million training images and $50,000$ validation images, where each image depicts natural scenes and has a resolution of $224 \times 224$. ImageNet is the de facto benchmark in image recognition in the previous decade. We also conduct pre-training on ImageNet-21K, a larger dataset of 21,841 classes (a superset of the 1,000 ImageNet-1K classes) with around 14M images.
- The MedMnist decathlon challenge (Yang et al., 2021) is a collection of $10$ different classification tasks, with each task corresponding to a different type of medical images, such as chest X-rays, mammograms, and images of the eye. The dataset provides a performance evaluation on medical image analysis. Each image has a resolution of $28 \times 28$, while MedMnist is diverse on data scale (from 100 to 100,000) and tasks (binary/multi-class, ordinal regression, and multi-label).

**Baseline methods**: The following details are used for each implementation method:

Table 6: Hyperparameter details for different experiments. The column natural scenes correspond to the experiments in Table 18.

| Hyperparameter | Natural scenes | Medical images |
|---|---|---|
| initial learning rate | $3 \cdot 10^{-4}$ | $3 \cdot 10^{-4}$ |
| warmup | ✓ | ✓ |
| weight decay | $10^{-5}$ | $10^{-5}$ |
| batch size | 256 | 512 |
| training epochs | 300 | 100 |
| scheduler | cosine | cosine |
| label smoothing | 0.15 | 0.05 |
| auto augmentation | ✓ | - |
| random erase | 0.1 | - |

- For the MLP-Mixer (Tolstikhin et al., 2021), we do not use the pretraining on ImageNet to report results on smaller datasets. Instead, we run the method directly on the training set for a fair comparison with other methods. We use a patch size of $4$ for images of the resolution of $32 \times 32$, i.e., in Cifar-10, Cifar-100, but we use the same data augmentation and training algorithms as in RPS.

- For SimplePatch (Thiry et al., 2021), we use the public repo to extend their results in datasets not reported.

- For $\Pi$-Nets (Chrysos et al., 2020), we report the results on polynomial networks without using activation functions. This was also reported in the original paper.

- For SENet (Hu et al., 2018) and ResNet (He et al., 2016), we utilize public repositories to report the accuracies, using a setup similar to $\Pi$-Nets.

**Hyperparameter details**: We provide the hyperparameter details used for the RPS in Table 6. We intend to release the source code upon the acceptance of the paper publicly, but the tables below can be a quick reference point for the core hyperparameters.

# B ABLATION STUDIES

Below we perform a range of additional experiments and provide further details for the experiments on the main paper. Concretely, in Appendix B.1, we include the additional tables supporting the experiments in the main table. Subsequently, we conduct a series of ablation studies on the initialization scheme (Appendix B.2), ordering of the different techniques (Appendix B.5). We confirm the improvement of each component on STL-10 in Appendix B.3, while we evaluate the runtime cost of each component in Appendix B.4. We conduct an extensive evaluation on diverse datasets and different networks in Appendix B.10.

## B.1 COMPLEMENTARY TABLES FOR EXPERIMENTS IN THE MAIN PAPER

The following tables below complement the corresponding experimental evaluation in the main paper:

- Table 7 complements the experiment in Sec. 3.2 in the main paper by offering the exact numerical values from Pi-Sigma to RPS.

- Table 8 offers the exact numerical values for the MedMNIST challenge of Sec. 3.6.

- Table 9 provides the numerical values for the experiment on limited data on Cifar-10 from Sec. 3.7.

As a reference we add the performance of ResNet, MLP-Mixer and Pi-Sigma using a single block with AdamW optimizer in Table 10.

Table 7: Improving Pi-Sigma step-by-step using each technique. All models are trained on a single GPU on Cifar-10 and Cifar-100. Experimental details in Sec. 3.2 (main paper). From left to right: the second column runs the original Pi-Sigma using SGD with fixed learning rate. The third column uses the original model, but learned with AdamW and with a scheduler. Then, each next column inserts one technique from the Sec. 2.

| Dataset | Eq. $(\Pi-\Sigma)$† | Eq. $(\Pi-\Sigma)$ | Technique 1 | Technique 2 | Technique 3 | Technique 4 | Technique 5 | Technique 6 | RPS ⋆ |
|---|---|---|---|---|---|---|---|---|---|
| Cifar-10 | 0.244 | 0.464 | 0.525 | 0.581 | 0.735 | 0.743 | 0.745 | 0.783 | 0.880 |
| Cifar-100 | 0.041 | 0.171 | 0.208 | 0.257 | 0.433 | 0.449 | 0.485 | 0.552 | 0.637 |

The '†' denotes the variant that is run with vanilla SGD.

The '⋆' denotes the final model, i.e., Technique 6, plus data augmentation and Label smoothing.

Table 8: Performance in MedMNIST challenge. The best are marked in **bold**. RPS performs favorably to ResNet in six out of eleven datasets. The original experimental description can be found in Sec. 3.6 (main paper).

| Dataset | ResNet-18 | RPS | Pi-Sigma |
|---|---|---|---|
| PathM | 0.907 | **0.914** | 0.355 |
| DermaM | 0.735 | **0.772** | 0.668 |
| OCTM | 0.743 | **0.782** | 0.278 |
| PneumoniaM | 0.854 | **0.918** | 0.625 |
| RetinaM | 0.524 | **0.557** | 0.435 |
| BreastM | **0.863** | 0.801 | 0.730 |
| BloodM | 0.930 | **0.966** | 0.198 |
| TissueM | **0.677** | 0.660 | 0.381 |
| OrganAM | **0.935** | 0.899 | 0.282 |
| OrganCM | **0.900** | 0.852 | 0.222 |
| OrganSM | **0.782** | 0.725 | 0.235 |

## B.2  ABLATION STUDY ON THE INITIALIZATION SCHEME

We conduct an ablation study on the standard initializations, including the popular Xavier (Glorot & Bengio, 2010) and He initializations (He et al., 2015). For this experiment, we do not utilize data augmentations, in order to avoid any confounding factors with the initialization performance. The results in Table 11 confirm that the popular He normal initialization performs slightly better than the rest. In the rest of the paper, we utilize the He normal initialization though owing to the simplicity of use. Therefore, the performance of RPS can be further augmented through further investigation of the impact of the initialization, which we defer to future work.

Table 9: Performance in Cifar-10 when trained with limited data. We compare our RPS with MLP-Mixer and Pi-Sigma. The original experimental description can be found in Sec. 3.7 (main paper).

| Sample numbers | MLP-Mixer | RPS | Pi-Sigma |
|---|---|---|---|
| 50 | 0.404 | 0.413 | 0.173 |
| 100 | 0.467 | 0.494 | 0.178 |
| 250 | 0.542 | 0.573 | 0.174 |
| 500 | 0.645 | 0.645 | 0.183 |
| 1000 | 0.696 | 0.695 | 0.206 |
| 1500 | 0.730 | 0.728 | 0.217 |
| 2000 | 0.754 | 0.750 | 0.224 |
| 3000 | 0.774 | 0.780 | 0.217 |
| 5000 | 0.826 | 0.805 | 0.225 |

Table 10: Performance of ResNet, MLP-Mixer and Pi-Sigma. This can be used as a baseline for the comparison. In each case, we use a single block using AdamW.

|  | Cifar-10 | Cifar-100 |
|---|---|---|
| ResNet (single block) | 0.825 | 0.545 |
| MLP-Mixer (single block) | 0.732 | 0.438 |
| Pi-Sigma (single block) | 0.464 | 0.171 |

Table 11: Ablation study on the initializations.

| **Initialization** | Xavier Uniform | He normal | Normal |
|---|---|---|---|
| **Acc.** | 0.811 | 0.816 | 0.806 |

### B.3 FROM PI-SIGMA TO RPS ON STL-10

We extend the experiment of Sec. 3.2 (main paper) to STL-10, which has images of higher resolution. The results in Fig. 4 verify that the learning algorithm and the normalization scheme are important. One point of differentiation from the previous results is that making the network deeper is important as a standalone technique, while in the rest of our experiments, depth enables the rest techniques to achieve higher performance.

### B.4 RUNTIME COST OF EACH COMPONENT

The empirical validation has established the improvement offered by each individual component, but a reasonable question is what the computational cost of each technique is. We analyze the throughput of the network when each technique is added progressively. We use Cifar-10 as the testbed here using a single GPU with a batch size of 64 and report the results in Table 12 and visualize them in Fig. 6. Notice that each technique results in a reduction of the number of images processed per second, which is expected. The largest impact is caused by Technique 1, which creates a deep network.

### B.5 FROM PI-SIGMA TO RPS: ABLATION STUDY ON THE TECHNIQUE ORDERING

As a reminder, the techniques have been applied with the prescribed order outlined in Sec. 2 so far. One reasonable question is whether a different ordering would change the outcomes or not. In this section, we consider alternative orderings in order to evaluate this impact.

In Table 13, we exhibit an alternative ordering that introduces each technique individually without the progressive addition of previous ideas outlined in Sec. 2. For example, in the fourth column, we solely incorporate Layer Normalization to a shallow model with a single block. The experimental results consistently validate our main findings, emphasizing the crucial role of the algorithm, normalization scheme, and data augmentations in achieving favorable outcomes. Notably, we observe that increasing the network depth positively affects performance, although this effect manifests at a later stage in this alternative ordering. Furthermore, the shallow model augmented with normalization and AdamW optimization exhibits promising results. Additionally, we highlight that the inclusion of the transpose block yields a substantial improvement in this particular context.

Table 12: Throughput of images from the network when different techniques are added. The throughput is tested on Cifar-10 images in a single GPU with batch size 64.

| Model | Pi-Sigma | Adam | Technique 1 | Technique 2 | Technique 3 | Technique 4 | Technique 5 | Technique 6 | RPS ⋆ |
|---|---|---|---|---|---|---|---|---|---|
| Throughput (image/sec) | 11299 | 11088 | 1317 | 1046 | 737 | 559 | 520 | 270 | 263 |

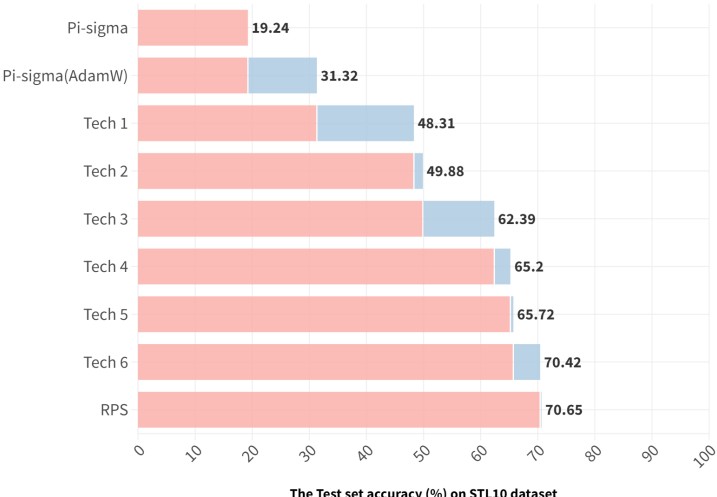

Figure 4: Step-by-step improvement of Pi-Sigma on STL-10. The blue color depicts the absolute improvement in accuracy when adding techniques incrementally. RPS is the final model, i.e., Technique 6, plus data augmentation and label smoothing. Notice that each additional technique has a positive contribution to the final performance. In fact, we notice that the critical elements are the learning algorithm (from first to second row), Technique 1 and Technique 3.

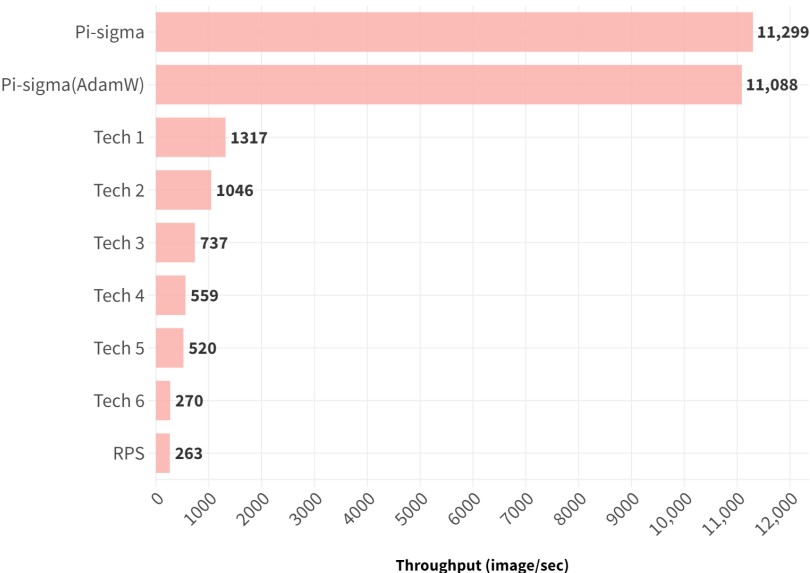

Figure 5: Throughput of images from the network when different techniques are added. The throughput is tested on Cifar-10 images in a single GPU with batch size 64.

Given the significance of few components, we consider one more ordering, where the critical techniques are used earlier. The results in Table 14 confirm that even under this scenario, the results remain similar.

**Shallow model**: An interesting question is how well the method performs when using a shallow model with one block. To this end, we utilize all the techniques, apart from Technique 1, i.e., we assume a shallow model with a single block. The final accuracy is 0.605 for Cifar-10 and 0.333 for

Table 13: Improving Pi-Sigma step-by-step using each technique in an alternative ordering to the main paper. All models are trained on a single GPU on Cifar-10 and Cifar-100. From left to right: the second column runs the original Pi-Sigma using SGD with a fixed learning rate. The third column uses the original model, but learned with AdamW and with a scheduler. Then, each next column inserts one technique from Sec. 2. *Differently to the previous tables, in this case, each column adds the idea of the technique in isolation* and not all the previous techniques that are progressively added in Sec. 2. That is, the fourth column adds only the Layer Normalization to a shallow model with a single block. The results confirm our main outcomes, i.e., the critical role of the algorithm, the normalization and the data augmentations in the end. Notice that making the network deeper (i.e. including Technique 1) improves the performance, but this technique emerges later in this ordering. The shallow model with the normalization and the AdamW performs well.

| Dataset | Eq. $(\Pi-\Sigma)$† | Eq. $(\Pi-\Sigma)$ | Technique 3 | Technique 5 | Technique 2 | Technique 1 | Technique 6 | Technique 4 | RPS ★ |
|---|---|---|---|---|---|---|---|---|---|
| Cifar-10 | 0.244 | 0.464 | 0.569 | 0.586 | 0.617 | 0.753 | 0.839 | 0.841 | 0.880 |
| Cifar-100 | 0.041 | 0.171 | 0.229 | 0.301 | 0.306 | 0.470 | 0.536 | 0.544 | 0.637 |

The '†' denotes the variant that is run with vanilla SGD.

The '★' denotes the final model with all the techniques added plus data augmentation and label smoothing.

Table 14: Improving Pi-Sigma step-by-step using each technique in an alternative ordering to the main paper. All models are trained on a single GPU on Cifar-10 and Cifar-100. From left to right: the second column runs the original Pi-Sigma using SGD with a fixed learning rate. The third column uses the original model, but learned with AdamW and with a scheduler. Then, each next column inserts one technique from Sec. 2. *Differently to the tables in the main paper, in this case, each column adds the technique in isolation* and not all the previous techniques that are progressively added in Sec. 2. That is, the fourth column adds only the Layer Normalization to a shallow model with a single block. The results confirm our main outcomes, i.e., the critical role of the algorithm, the normalization and the data augmentations in the end.

| Dataset | Eq. $(\Pi-\Sigma)$† | Eq. $(\Pi-\Sigma)$ | Technique 3 | Technique 2 | Technique 1 | Technique 6 | Data augm ★ | Technique 4 | Technique 5 |
|---|---|---|---|---|---|---|---|---|---|
| Cifar-10 | 0.244 | 0.464 | 0.569 | 0.601 | 0.724 | 0.724 | 0.842 | 0.861 | 0.880 |

The '†' denotes the variant that is run with vanilla SGD.

The '★' denotes the addition of data augmentation and label smoothing.

Cifar-100. The accuracy is significantly decreased from the final accuracy of Table 7, which verifies our hypothesis that depth is an enabler for other techniques to improve the accuracy.

## B.6 ADDING ONE TECHNIQUE AT A TIME

Our last experiment with the shallow model poses the question of what the impact of each *individual* technique is when added in isolation and not incrementally. We conduct the experiment on Cifar-10 and report the results in Table 15.

## B.7 IMPACT OF THE SKIP CONNECTION

An interesting question is whether we can remove the skip connection. A naive implementation of directly removing the skip connection fails, which is precisely why we pair the skip connection with the deep number of layers in Technique 1. Instead, we add a scalar $\rho$ to scale the influence of the skip connections across the network. The $\rho$ is sampled from a Gaussian distribution with a mean $\mu$ and a variance of $10^{-5}$. The mean starts at 1 in the beginning of the training, which means that we use the regular (unscaled) skip connection and then the mean decreases during the training, therefore decreasing the contribution of the skip connection. Interestingly, the accuracy of the network seems

Table 15: Ablation on adding each technique *in isolation*, i.e., on top of the original Eq. ($\Pi-\Sigma$) when run with AdamW. The first two columns of results report the baseline performance when trained with SGD and AdamW for reference.

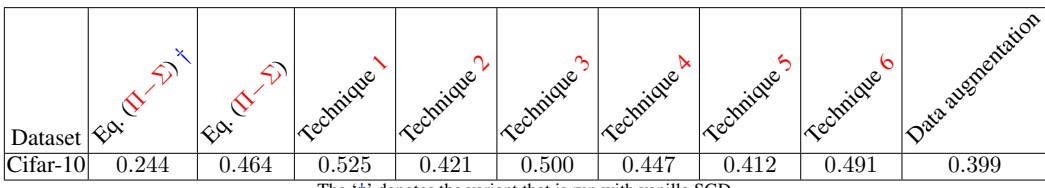

| Dataset | Eq. ($\Pi-\Sigma$)$^\dagger$ | Eq. ($\Pi-\Sigma$) | Technique 1 | Technique 2 | Technique 3 | Technique 4 | Technique 5 | Technique 6 | Data augmentation |
|---|---|---|---|---|---|---|---|---|---|
| Cifar-10 | 0.244 | 0.464 | 0.525 | 0.421 | 0.500 | 0.447 | 0.412 | 0.491 | 0.399 |

The '$\dagger$' denotes the variant that is run with vanilla SGD.

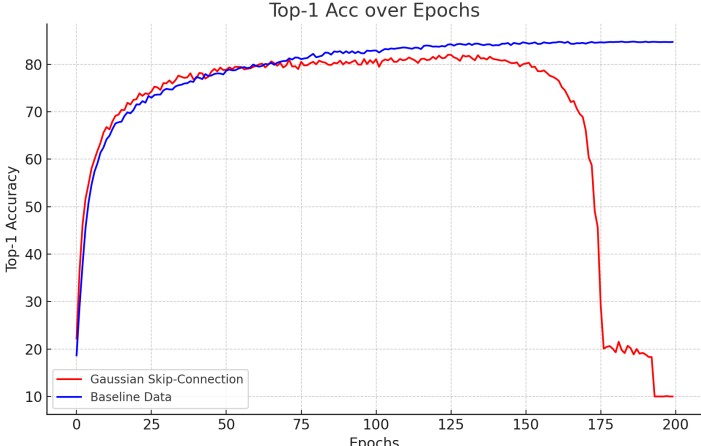

Figure 6: Ablation study on scaling the skip connection and diminishing its impact throughout the training. The experiment is performed on Cifar-10 and it compares the RPS with the variant of scaled skip connections. For the first 75 epochs, the performance of the two networks is similar. Then, the dramatic change occurs around epoch 150.

to be increasing up to the point that the mean value becomes $0.04$ as illustrated in Fig. 6. It is possible that a more refined analysis could further increase the performance of the network and/or completely remove the skip connections, however this is not our main goal.

## B.8 Impact of the optimizer

We conduct an ablation study focused on the role of the optimizer. We leave all the rest components in-tact and simply change the optimizer to assess the performance when different optimizers are used. The results in Table 16 exhibit that Adam variants work well, but SGD does not. Surprisingly, the older optimizer of RMSProp also works well.

## B.9 Impact of the data augmentation technique

We assess the impact of removing a single data augmentation technique at a time, while the rest of the components remain in-tact. The results in Table 17 indicate that the techniques have a similar role and there is a drop when one of them is not used.

Table 16: Ablation study on the optimizer used in RPS when trained on Cifar-10.

| Optimizer | AdamW | Adam | AdamP | SGD | Lars | RMSProp |
|---|---|---|---|---|---|---|
| Accuracy | 0.880 | 0.845 | 0.847 | 0.375 | 0.294 | 0.878 |

Table 17: Ablation study on the data augmentation techniques used in RPS when trained on Cifar-10.

|  | No Auto augment | No Mixup | No smoothing | No random erase | Baseline |
|---|---|---|---|---|---|
| Accuracy | 0.860 | 0.846 | 0.854 | 0.854 | 0.880 |

### B.10 COMPARISON WITH DIVERSE NETWORKS IN SIX DATASETS ON IMAGE RECOGNITION

Below, we conduct experiments with seven classical datasets, which are split into two categories: the first includes images of resolution up to $100 \times 100$, and the other category includes the challenging ImageNet.

**Baselines**: Seven methods are used for comparison. We utilize ResNet-18 as the default baseline in recognition. We also utilize the popular baseline of SENet (Hu et al., 2018). Similarly to the Pi-Sigma that captures high-order correlations, Π-Nets (Chrysos et al., 2020) are utilized as a baseline, since they also can capture high-order correlations. We also add indicative architectures used in the previous decade for recognition, such as NiN (Lin et al., 2014), SCKN (Mairal, 2016), SimplePatch (Thiry et al., 2021), MLP-Mixer (Tolstikhin et al., 2021). For each baseline, we report the accuracy reported in the original paper. We augment those results by executing the baseline methods in additional datasets using public repositories. Our goal is to position the revised Pi-Sigma with respect to recent methods and not to exhaustively try every additional regularization method in each baseline. As such, we use the compared methods relying on open-source implementations.

In Table 18, we report the results[2] on six datasets (i.e., Cifar-10, Cifar-100, STL-10, Tiny ImageNet, O-Flowers) along with the number of parameters and the FLOPs of each method. The FLOPs are measured on Cifar-10. Evidently, the dated Pi-Sigma fares unfavorably compared to contemporary deep learning methodologies, a consequence befitting its vintage. Nonetheless, we put forth evidence that employing straightforward techniques to Pi-Sigma can lead to a drastic enhancement of up to $2500\%$ in performance. A significant rise in performance is observed for Cifar-100, where the original block proves to be an inadequate fit for 100 classes. Despite RPS demonstrating comparable results to formidable baselines such as SCKN, Π-Nets, MLP-Mixer and SimplePatch on most datasets, stronger baselines surpass RPS. To the credit of RPS, it performs equivalently with the leading baselines on SVHN. Our findings serve to emphasize the efficacy of common techniques in elevating the performance of Pi-Sigma, although it may not suffice in achieving state-of-the-art results with this fundamental architectural block.

## C IMPACT OF THE CONFIDENCE OF THE CLASSIFIER

Beyond the improvement of the accuracy, a key question in deep learning is the confidence of the network with respect to its prediction. To our knowledge, there are few to no insights with respect to the impact of different components on the confidence of the classifier. We conduct an experiment below to uncover the effect of each component on the confidence.

The experiment is designed as follows: We train the networks using the different techniques on Cifar-10 as described in Sec. 3.2 and utilize the pre-trained networks. We utilize only the datapoints that have a correct prediction and obtain the confidence of the classifier (i.e., the output of the softmax) for each datapoint. Then, we simply plot the distribution as quantized in 10 bins. The results in Fig. 7 exhibit the following interesting patterns: (a) layer normalization has a major impact on the confidence of the classifier, while (b) Technique 6 also has a major impact on the confidence. The impact of the transpose block can be attributed to the sharing of information across tokens through the block.

Beyond the aforementioned question, we also extend our confidence evaluation in the case of classification under noisy settings. We conduct the experiment on Cifar-10-C. Concretely, we use the Gaussian noise corruption with 5 corruption levels. For further details on the dataset, please check out Appendix D.

---

[2]Each experiment we conduct is repeated three times and we report the mean in the table to avoid cluttering the results. Table 19 reports the variance of each method.

Table 18: Accuracy of different architectures on six datasets. The last two columns report the FLOPs and the number of parameters (in millions) respectively. The last row displays the relative improvement from the original Pi-Sigma to the revised one (RPS). Notice that the RPS improves substantially over the original Pi-Sigma block. Importantly, RPS performs on par with a number of baselines (i.e., SCKN, Π-Nets, MLP-Mixer and SimplePatch).

| Method | Year | Cifar-10 | Cifar-100 | STL-10 | SVHN | Tiny ImageNet | O-Flowers | FLOPs | # Par |
|---|---|---|---|---|---|---|---|---|---|
| NiN | 2014 | 0.920 | 0.643 | - | - | - | - | - | - |
| SCKN | 2016 | 0.895 | 0.610 | 0.527 | - | 0.409 | - | - | 3.4 |
| ResNet-18 | 2016 | 0.944 | 0.756 | 0.741 | 0.961 | 0.615 | 0.877 | 0.56 | 11.7 |
| SENet | 2018 | 0.946 | 0.760 | 0.752 | 0.962 | | - | - | 11.6 |
| Π-Nets | 2020 | 0.907 | 0.677 | 0.563 | 0.961 | 0.502 | 0.826 | 0.59 | 11.9 |
| SimplePatch | 2021 | 0.885 | 0.652 | 0.694 | 0.876 | - | - | - | 85.5 |
| MLP-Mixer | 2021 | 0.907 | 0.649 | 0.712 | 0.968 | 0.480 | 0.906 | 1.21 | 12.3 |
| Pi-Sigma (single block) | 1991 | 0.244 | 0.041 | 0.213 | 0.196 | 0.019 | 0.070 | 0.03 | 0.60 |
| RPS (ours) | | 0.880 | 0.637 | 0.731 | 0.971 | 0.500 | 0.786 | 0.38 | 11.4 |
| **Improvement** (%) | | 260.7% | 1453.7% | 243.2% | 395.4% | 2531.6% | 1022.9% | | |

Coloring denotes the category of the method: ■ CNN-based ■ MLP-based

Table 19: Accuracy (mean and variance) of different architectures on six datasets. This table is complementary to the Table 18 in the main paper to demonstrate the variance. We report only the methods that have multiple runs here. The last row displays the relative improvement from the original Pi-Sigma to the revised one (RPS).

| Method | Year | Cifar-10 | Cifar-100 | STL-10 | SVHN | Tiny ImageNet | O-Flowers |
|---|---|---|---|---|---|---|---|
| SCKN | 2016 | $0.895 \pm 0.002$ | $0.610 \pm 0.002$ | $0.527 \pm 0.012$ | - | $0.409 \pm 0.001$ | - |
| ResNet-18 | 2016 | $0.944 \pm 0.001$ | $0.756 \pm 0.003$ | $0.741 \pm 0.016$ | $0.961 \pm 0.001$ | $0.615 \pm 0.002$ | $0.877 \pm 0.013$ |
| SENet | 2018 | $0.946 \pm 0.001$ | $0.760 \pm 0.003$ | $0.752 \pm 0.011$ | $0.962 \pm 0.001$ | - | - |
| Π-Nets | 2020 | $0.907 \pm 0.003$ | $0.677 \pm 0.006$ | $0.563 \pm 0.008$ | $0.961 \pm 0.003$ | $0.502 \pm 0.007$ | $0.826 \pm 0.001$ |
| SimplePatch | 2021 | $0.885 \pm 0.001$ | $0.652 \pm 0.002$ | $0.694 \pm 0.004$ | $0.876 \pm 0.001$ | - | - |
| Pi-Sigma (single block) | 1991 | $0.244 \pm 0.005$ | $0.041 \pm 0.001$ | $0.213 \pm 0.005$ | $0.196 \pm 0.001$ | $0.019 \pm 0.005$ | $0.070 \pm 0.001$ |
| RPS (ours) | | $0.880 \pm 0.012$ | $0.637 \pm 0.020$ | $0.731 \pm 0.005$ | $0.971 \pm 0.001$ | $0.500 \pm 0.001$ | $0.786 \pm 0.002$ |
| **Improvement** (%) | | 260.7% | 1453.7% | 243.2% | 395.4% | 2531.6% | 1022.9% |

Coloring denotes the category of the method: ■ CNN-based ■ MLP-based

In Fig. 8, we observe that the changes in the case of the robustness are more mild than in the case of Fig. 7. For instance, notice that the Technique 3 has a significantly reduced impact on the classifier confidence in this case.

# D  ROBUSTNESS

One interesting aspect beyond the accuracy is the robustness to diverse perturbations. Let us evaluate the robustness of RPS. The ImageNet-A dataset (Hendrycks et al., 2021) contains 7,500 natural adversarial examples of 200 classes from ImageNet-1K. The metric for assessing robustness to adversarially filtered examples for classifiers is the top-1 accuracy on ImageNet-A. The ImageNet-C dataset (Hendrycks & Dietterich, 2019) consists of 15 types of generated corruptions from noise, blur, weather, and digital categories. Each type of corruption has five levels of severity. Overall, the ImageNet-C dataset consists of 75 corruptions, all applied to ImageNet validation images for testing a pre-existing network. Here, we follow Hendrycks & Dietterich (2019) to aggregate the classifier's

Table 20: Comparison of image classifier robustness. The mCE value is the mean Corruption Error of the corruptions of Noise, Blur, Weather, and Digital.

| Model | ImageNet-1K (Acc %) ↑ | ImageNet-A (Acc %) ↑ | ImageNet-C (mCE %) ↓ |
|---|---|---|---|
| ResNet-18 | 69.90 | 1.15 | 84.7 |
| RPS ($o = 128$, $L = 51$) Small | 64.31 | 9.42 | 78.6 |

Table 21: Improving Li (2003) step-by-step using each technique. All models are trained on a single GPU on Cifar-10 and Cifar-100. From left to right: the second column runs the original model using SGD with fixed learning rate. The third column uses the original model, but learned with AdamW and with a scheduler. Then, each next column inserts one technique from the Sec. 2, but on top of the Li (2003) model. Each result is an average over three runs.

| Dataset | Original † | Original | Technique 1 | Technique 2 | Technique 3 | Technique 4 | Technique 5 | Technique 6 | Final ⋆ |
|---|---|---|---|---|---|---|---|---|---|
| Cifar-10 | 0.193 | 0.530 | 0.544 | 0.546 | 0.720 | 0.733 | 0.756 | 0.837 | 0.871 |
| Cifar-100 | 0.023 | 0.168 | 0.227 | 0.239 | 0.436 | 0.454 | 0.480 | 0.526 | 0.543 |

The '†' denotes the variant that is run with vanilla SGD.

The '⋆' denotes the final model, i.e., Technique 6, plus data augmentation and Label smoothing.

performance across severities and corruption types. Even though our small model does not perform favorably to ResNet-18 on ImageNet-1K, RPS outperforms ResNet on ImageNet-A and ImageNet-C, indicating an improved robustness of the proposed method.

# E    EXTENSION BEYOND THE PI-SIGMA

Even though confirming the validity of the proposed technique in every architecture is impossible, we scrutinize the validity of those components beyond Pi-Sigma. Concretely, Li (2003) is selected as the original model. Table 21 reports the results in adding each technique sequentially. The training algorithm and the normalization are still two critical techniques, with their impact being even more pronounced in this case.

# F    BROADER IMPACT AND ETHICS STATEMENT

**Broader impact**: This work does not aim to produce state-of-the-art results, but rather to evaluate a thirty year-old architecture in image recognition. Since we do not have any state-of-the-art results, we do not expect any negative societal bias from our method. We do encourage the community though to study further the broader impact of all types of models.

**Ethics statement**: We do not conduct any human studies or studies that involve animals in this work. We also do not release any dataset, we use standard, publicly-available datasets.

# G    TECHNIQUES WITH NEGATIVE RESULTS

We strive to offer an all-inclusive account of our efforts to enhance the performance of our approach, which includes reporting on techniques that were unsuccessful in yielding improvements. Although these results did not lead to the desired outcome, we believe they hold value in providing a more comprehensive understanding of our approach. Note that these negative results are specific to our experimental setup and may not extend to other problems or implementations. Therefore, we advise against drawing broad conclusions from negative results, as it may curtail potentially fruitful research directions.

- We found that *dropout* in the final layer might lead to deteriorated performance.

- *Dropout* in the intermediate layers never improved the performance, but often led to decreased accuracy.
- *Dropblock* did not provide any benefit in our case.
- Changing the sigmoid to another activation function, did not lead to any improvement.
- We tried increasing $N$ beyond $5$, but this led to a decreased performance. The symbol $N$ measures how many representations are multiplied together in Eq. ($\Pi-\Sigma$). We believe it is interesting to explore further techniques that expand on $N$ beyond $5$, implying higher-order correlations are captured inside each block.

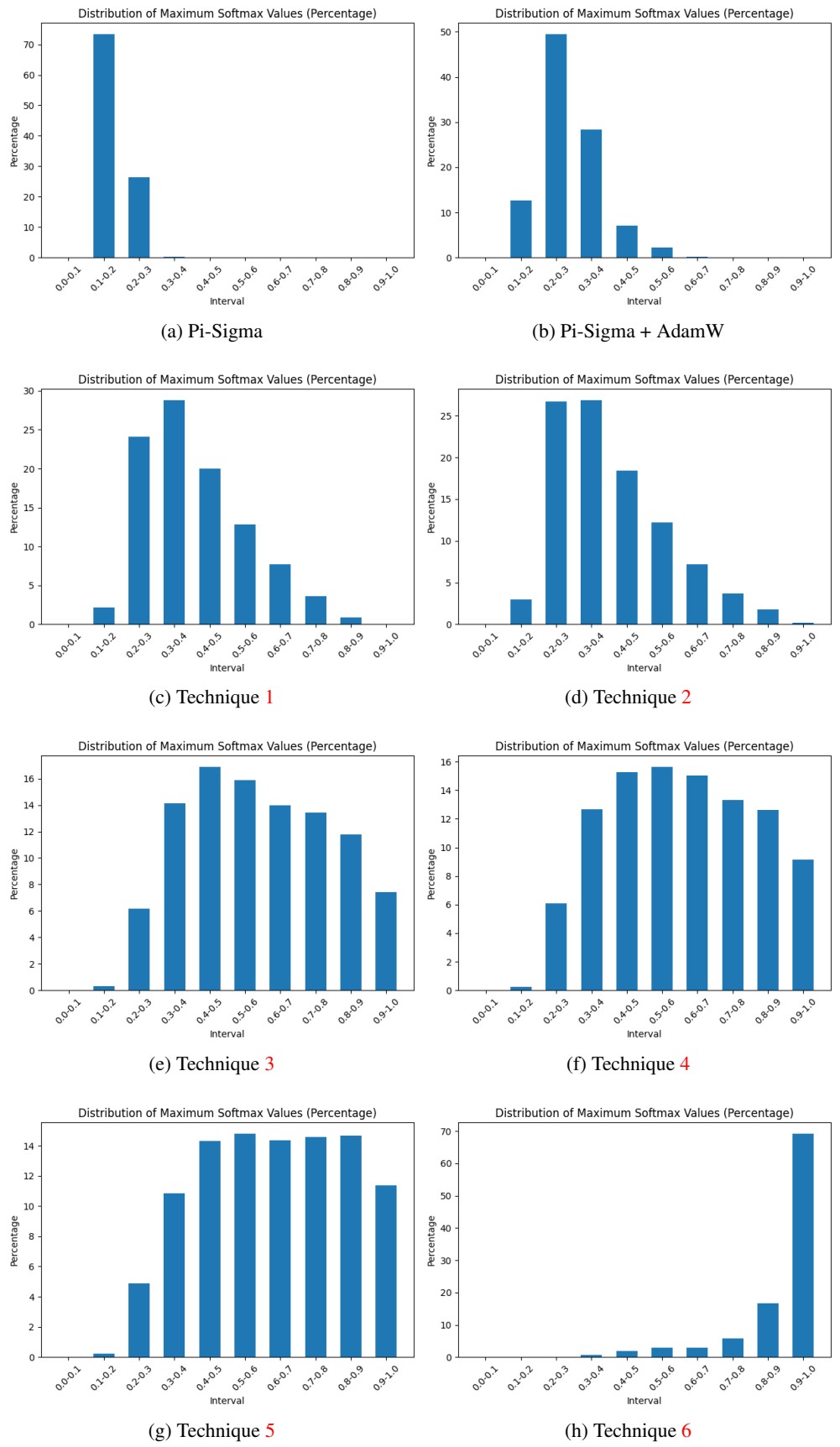

Figure 7: Confidence of the classifier when we add techniques one-by-one as developed in Sec. 3.2. The x-axis measures the confidence of the classifier (as measured by the softmax), while the y-axis reflects the frequency of occurrences over the dataset. The precise details of the experiment are described in Appendix C.

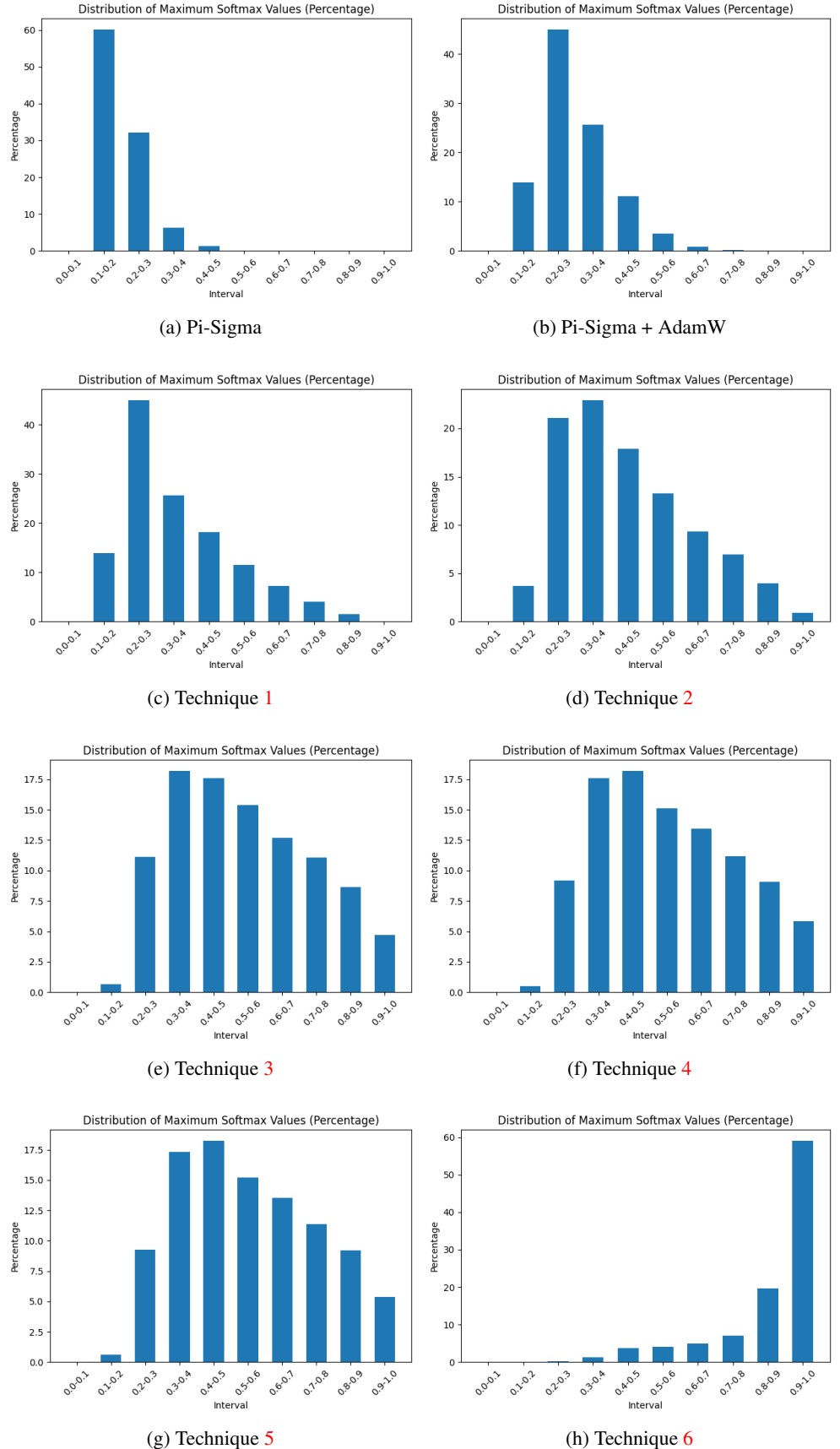

Figure 8: Confidence of the classifier (under a noisy setting) when we add techniques one-by-one as developed in Sec. 3.2. The x-axis measures the confidence of the classifier (as measured by the softmax), while the y-axis reflects the frequency of occurrences over the dataset. In contrast to Fig. 7, the changes in the confidence here are smaller as we add more techniques.

