# OpenReview forum: "An old dog can learn (some) new tricks: A tale of a three-decade old architecture"
_ICLR.cc/2024/Conference — Submitted to ICLR 2024_

### Official Review · Reviewer_AAyP · 2023-10-29

**Soundness:** 2 fair
**Presentation:** 3 good
**Contribution:** 1 poor
**Rating:** 3
**Confidence:** 4

**Summary:**

In this paper, the authors study the Pi-Sigma architecture proposed in 1991. The major objective is to see if this "old architecture" can be effectively improved by several modern techniques, e.g., skip connection, improved training algorithms, normalization layers, and data augmentation. The authors find that while an ensemble of ingredients bears significance in achieving commendable performance, only a few pivotal components have large impacts.

**Strengths:**

1. The writing of this paper is well.
2. The experiments are comprehensive in terms of studying existing techniques proposed in recent years.

**Weaknesses:**

My major concern lies in the significance of the contributions of this paper:

1. All the techniques studied in this paper have been proposed in existing works.
2. The effectiveness of these techniques has been confirmed in existing works.
3. Even with all the state-of-the-art techniques, Pi-Sigma net cannot outperform even ResNet-50 (with the same FLOPs, see: Table 4). The difference is ~5%, which is extremely large on ImageNet.
4. Even though this paper reveals that some techniques are less effective on Pi-Sigma, it may be hard to say that these techniques are not effective on top of other models.

**Questions:**

See weaknesses.

---

> ### Author Response · Authors · 2023-11-16
> **Response to Reviewer AAyP**
>
> We are thankful to the reviewer AAyP for their effort in reviewing our work. We address their concerns below:
>
> > **Q1**: The techniques have been previously proposed.
>
> We do agree with the reviewer. We believe that a retrospective view of existing techniques is valuable and can offer new insights that can be useful to practitioners. For instance, please check the [response to Q1 from reviewer jSdE](https://openreview.net/forum?id=yqAToOgxgf&noteId=qcNLj5PywS) on the surprising components on ViT and MLP-Mixer.
> ______
>
> > **Q2**: The effectiveness of these techniques has been confirmed by existing works.
>
> We agree that many of the techniques studied in this work have been popular in the literature. However, there is a *gap between knowledge of useful components and understanding when to use them and whether they would work in other models*. To our knowledge, many of these techniques are used exclusively in contemporary works, making the difference from older models more opaque. Concretely, is there any paper showing that the dated Pi-Sigma (a network proposed over 30 years ago) can perform so well by the addition of those standard techniques? If the reviewer has seen such a paper previously, please let us know and we will adjust our claims.
> ______
>
> > **Q3**: Pi-Sigma net cannot outperform even ResNet-50.
>
> Let us clarify that our large model does perform favorably to the ResNet50 (given more FLOPs though), please check Table 4. However, the goal of this work is not to claim state-of-the-art behavior. We would like to remind the reviewer that when MLP-Mixer firstly emerged, it could not surpass ResNets. Fortunately though, several brave researchers continued their dedicated effort to improve upon those models and propose improvements on those. That resulted in important improvements in the model design of MLP-based models and has led to several successful models since. Therefore, we do urge the reviewer to consider our contributions as a retrospective analysis that includes several fresh insights offered in the paper.
> ______
>
> > **Q4**: It may be hard to say that these techniques are not effective on other models.
>
> We agree with the reviewer. We have made an effort to extend beyond Pi-Sigma, please check sec. D in the Appendix. Having said that, we are open to consider an alternative network if the reviewer has any suggestion for us.

---

### Official Review · Reviewer_6g4Q · 2023-10-30

**Soundness:** 4 excellent
**Presentation:** 3 good
**Contribution:** 2 fair
**Rating:** 5
**Confidence:** 3

**Summary:**

This paper conducts a comprehensive analysis of Pi-Sigma and argues that there is an abundance of techniques available in the last few years and yet few insights into how each technique fares in other architectures or its role in generalization in new datasets. It confirms experimentally that, through the strategy of established techniques, the refined architecture achieves performance levels comparable to recent architectures. This work facilitates a comprehensive understanding of the benefits derived from a holistic evaluation of models, shedding light on the key techniques that drive architectural advancements.

**Strengths:**

1. The author conducted an extensive range of experiments in this article to provide insightful guidance for future model design.

2. The logical structure of this article is clear and makes it easy for readers to follow.

3. In this article, the author presents an interesting discovery where an older model can achieve state-of-the-art performance by incorporating some existing techniques.

**Weaknesses:**

1. The motivation for this article is insufficient. The author aims to provide insights for future model design through experiments, but these findings are all based on the Pi-Sigma model, which may not necessarily apply to future model designs and could even be misleading.

2. The experiments conducted in this article exclusively employ existing techniques, failing to offer guiding principles for the innovation of new methods in the future.

3. ViT is currently a widely used model, and it is known for its scalability on data. However, despite the experiments conducted on ImageNet-21k, the author did not include comparative experiments or discussions regarding the ViT model.

4. The author discusses the impact of adding six different techniques to the Pi-Sigma model in various orders. However, in the ablation experiments section, there is no discussion of the individual effects of adding each technique to the Pi-Sigma model separately.

**Questions:**

1. Could the author discuss some guiding insights from this research for future model design, particularly for state-of-the-art CNNs and ViTs?

2. Could the author explain why they chose to use Pi-Sigma and highlight any unique aspects or clear advantages it has over CNNs and ViTs?

---

> ### Author Response · Authors · 2023-11-17
> **Response to Reviewer 6g4Q (1/2)**
>
> We are thankful to the reviewer 6g4Q for their effort and time to review our work. We appreciate their praise on the extensive experiments and the observations on older models. We address their concerns below:
>
> > **Q1**: Findings based on the Pi-Sigma model and might not reflect other models.
>
> We agree with the reviewer that *any* empirical analysis has no guarantees over the design of other models. However, we advocate that our retrospective analysis and insights add a unique value, because our analysis assesses components proposed within the last few years in a model proposed 30 years ago. Therefore, the model is not "tuned" to work well with those components specifically. On the contrary, we offer a detailed account of what works, how much it improves the result and even what does not work.
> ______
>
> > **Q2**: Guiding principles for future model design
>
> We do believe that our insights can be useful to a practitioner in the field:
>
> * We do offer an analysis over the trade-offs of components (e.g. accuracy improvement over computational cost of adding them). The cost of those components is less dependent on the architecture.
>
> * Many of the insights offered in this work are not mentioned in the literature to our knowledge. For instance, the fact that the contribution of each technique depends considerably on the techniques added already. If the reviewer is aware of any previous reference or study in this, we would be happy to include it.
>
> * Last but not least, the role of skip connections. Even though skip connections are an integral part of model design, their exact role is unclear upon our study. Many works, including ViT, MLP-Mixers are using skip connections by default and do not elaborate on *why* those are included or their role.
>
> We do believe that those insights along with our findings of reviving an older architecture can have an impact in the community.
> ______
>
> > **Q3**: Include the ViT model.
>
> We appreciate the remark by the reviewer. We include ViT results in Table 4. Notice that ViT has both more FLOPs and more parameters than our largest model. Additionally, note that the best ViT result reported in the corresponding paper is pre-trained on JFT-300M, which is a proprietary dataset, so a direct comparison with models trained on public datasets might not be fair. We can also try to conduct experiments on the rest of the datasets for a fair comparison. We will update the tables when we have additional results.
> ______
>
> > **Q4**: Include the individual effects of adding each technique to the Pi-Sigma model separately.
>
> We are thankful to the reviewer for proposing this important experiment. We have indeed conducted this experiment, where techniques are added in isolation on top of the original Pi-Sigma block trained on AdamW. The results below report the average accuracy over 3 runs (on Cifar10):
>
>
> |      	| T. 1 | T. 2 | T. 3 | T. 4 | T. 5 | T. 6 | Data augm. |
> |----------|------|------|------|------|------|------|------------|
> | Accuracy | 52.5 | 42.1 | 50.0 | 44.7 | 41.2 | 49.1 | 39.9   	|
>
> Note that the strongest techniques of Layer Normalization and Deep Network work well in isolation (when trained with AdamW). This is complementary to our key observation that the ordering of adding components can play a key role in the additional performance of the network, as we emphasize in sec. 3.8.
>
> To provide a more thorough analysis, we also test the components in isolation using SGD as the learning algorithm and the results are exhibited below:
>
> |      	| T. 1 | T. 2 | T. 3 | T. 4 | T. 5 | T. 6 | Data augm. |
> |----------|------|------|------|------|------|------|------------|
> | Accuracy | 28.0 | 20.8 | 32.1 | 18.1 | 19.5 | 19.6 | 20.0   |
>
> This experiment has been added in sec. B.6 and highlighted with red in the revised manuscript for visual distinction.
> ______

---

> > ### Comment · Reviewer_6g4Q · 2023-12-01
> > **Response to the rebuttal**
> >
> > Thank the authors for providing a detailed rebuttal. Most of my concerns are well addressed.
> >
> > However, I am still not fully convinced by the contributions.
> >
> > The title of this work is 'An old dog can learn (some) new tricks: A tale of a three-decade old architecture'.
> >
> > The claimed contribution is 'Our work aims to uncover key insights on the critical techniques contributing to the exceptional performance of DNNs.'
> >
> > If the goal is to uncover key insights on the techniques critical to DNNs, why only Pi-sigma is chosen? It would be more convincing if the authors could draw similar conclusions and insights across different old-dog architectures.
> >
> > Hence, I tend to keep my original score. However, I lower my confidence. It is entirely acceptable to me if AC decides to accept this paper.

---

> ### Author Response · Authors · 2023-11-17
> **Response to Reviewer 6g4Q (2/2)**
>
> > **Q4**: Unique aspects of Pi-Sigma over CNNs and ViTs.
>
> Our experiments exhibit a concrete advantage of the revised Pi-Sigma when compared to contemporary models. The experiments in sec. C indicate that the proposed revised Pi-Sigma has significant benefits with respect to robustness in ImageNet-A and ImageNet-C. Robustness is a topic of ever-increasing significance as neural networks are deployed in real-world applications. At the same time, the understanding of robustness is not yet complete with even core insights such as depth or width in feedforward networks being unclear in all learning regimes [1]. Therefore, we do believe that this benefit of the proposed model might result in further insights into the robustness of less traditional models.
>
> Beyond the aforementioned advantage, let us emphasize that the goal of this work is not to design a network with a certain advantage, but rather to highlight how dated architectures can be "resurrected" using very standard components. Pi-Sigma is a prominent example of those architectures and in fact Pi-Sigma is mentioned in standard textbooks [2] as an example of a multilayer network, therefore making it an abandoned but known to the community usecase.
>
>
> # # References
>
> [1] Zhu, et al. “Robustness in deep learning: The good (width), the bad (depth), and the ugly (initialization)”, NeurIPS’22.
>
> [2] Mehrotra, et al. “Elements of artificial neural networks”. MIT press, 1997.

---

### Official Review · Reviewer_Gb9t · 2023-11-01

**Soundness:** 3 good
**Presentation:** 3 good
**Contribution:** 2 fair
**Rating:** 6
**Confidence:** 3

**Summary:**

This paper presents how they rejuvenated a 30-year-old architecture, $Pi-Sigma
$, with modern tools and techniques. The authors aim to investigate whether this old architecture can compete with contemporary models when equipped with modern tools. The paper systematically introduces a series of techniques to enhance the original $Pi-Sigma$ model, including the incorporation of skip connections, normalization schemes, data augmentation, and more recent technqiues.

**Strengths:**

- I really like the way the authors describe the premise of this paper and motivate their interests: find exactly what components of models cause the biggest impact on performance and this area got especially popular after the release [1, 2, 3] among others. The premise that the authors explore and the way they motivate the paper is really interesting.
- The authors very clearly identify the novelty in this work, "However, our objective is to delve deeper into architecture design and explore how
previously used techniques and tools"
- The authors try out quite a few diverse modern techniques and see how they impact performance on $\Pi-\Sigma$ network.
- I think the results and experiments are well put together, the authors not only show that they improve $\Pi-\Sigma$ networks but also extend their experiments to multiple datasets and also on $\Sigma\Pi\Sigma$ neural networks.

[1] Tolstikhin, Ilya O., et al. "Mlp-mixer: An all-mlp architecture for vision." Advances in neural information processing systems 34 (2021): 24261-24272.

[2] Smith, Samuel L., et al. "ConvNets Match Vision Transformers at Scale." arXiv preprint arXiv:2310.16764 (2023).

[3] Trockman, Asher, and J. Zico Kolter. "Patches are all you need?." arXiv preprint arXiv:2201.09792 (2022).

**Weaknesses:**

- Though the authors identify what training processes work well for neural net architectures and also show that they reach the performance of ResNet just due to these techniques, they do not share the kind of techniques that were proven to be highly architecture-specific and probably do not work for this. In general, I believe that the insights the authors came up with are not fully new and were known. I do not say that these kinds of insights are not valuable, but for instance, showing that patching has a big impact on performance, which was not known earlier.
- I think the authors only try out fairly popular techniques for the experiments, for which each of their individual effects is fairly well-studied and well-known. It would have been a much more interesting paper if the authors also included other low-level techniques that such models could try out.
- In this case the paper method might not come across as novel, one would often look towards the paper for using something in a new context, which the authors do pretty well and also build the RPS model or for some important insights which I believe this paper does not correctly do.

**Questions:**

- I believe a very important question is to understand what kind of techniques do not seem to improve performance, it would be of great interest to understand what kind of techniques are possibly over-reliant or dependent on certain architectures. I would also encourage the authors to include this important detail in their work.
- A typo in

> Out work aims

---

> ### Author Response · Authors · 2023-11-17
> **Response to Reviewer Gb9t**
>
> We are grateful to the reviewer Gb9t for their feedback and their appreciation of the novelty and interest of our work. We address their concerns below. The resulting changes in the paper are highlighted with red color for visual distinction. If there are any remaining unclear points, we would be happy to discuss them further.
>
> > **Q1**: For each of the techniques the individual effects are well-studied and well-known. It would have been a much more interesting paper if the authors also included other low-level techniques that such models could try out.
>
>
> We agree with the reviewer that these are popular techniques, we also mention this explicitly in our paper as well. However, we do not share the belief that their effects are well-known. For instance, despite the widespread adoption of skip connection, its role is still not conclusively understood as we also mention in sec. 3.8. In addition, in our [response in Q2 for reviewer jSdE](https://openreview.net/forum?id=yqAToOgxgf&noteId=qcNLj5PywS), we showcase how we could reduce the impact of the skip connection in the network.
>
> We advocate that there are fresh insights even in existing components: for instance, that the contribution of each technique depends considerably on the techniques added already. If the reviewer is aware of any previous reference or study in this, we would be happy to include it. Why is this significant? Practitioners that design new models can be directly impacted by this observation.
>
> Regarding additional techniques to try out, we also agree with the reviewer. **We are open to suggestions on techniques to try out** and explore their effect on our framework.
> ________
>
> > **Q2**: What kind of techniques do not improve the performance?
>
> We are thankful for the question. We do agree that the list of techniques with negative results is important. In fact, a list of the techniques that we tried and failed to provide any improvement exists in sec. F. Surprisingly, techniques once popular, e.g., dropout or dropblock, fail to provide any improvement in our case. We do believe that this list can be further augmented, we are willing to try out suggested techniques.
> _____
>
> > **Q3**: Typo in “Out work aims”.
>
> We are thankful to the reviewer for the attentive study of our work. The typo has been fixed.
>
> If the reviewer has any remaining concerns, we would be happy to clarify.

---

### Official Review · Reviewer_jSdE · 2023-11-01

**Soundness:** 3 good
**Presentation:** 3 good
**Contribution:** 3 good
**Rating:** 6
**Confidence:** 3

**Summary:**

The paper proposes to analyse various components in neural net training by building on top of a 30 year old architecture Pi-Sigma. Instead of coming up with novel components they propose to use previously used techniques to add it to the architecture.  After a lot of ablations they conclude that the most important components towards improvement are the i) optimizer ii) normalization iii) data augmentation and iv) depth of the neural net

**Strengths:**

- The idea of using an old architecture and building on top of it is unique and interesting.
- The paper does a good job at analyzing various components by building on top of an old architecture.
- The paper does a very dense evaluation to analyze a lot of components in the neural net training by testing over multiple datasets and having various architectures as baselines.
- The paper is well written.

**Weaknesses:**

- Although the analysis is useful to put it out there, i'm not sure if it was surprising at the end. As the current architectures of MLP mixer or ViT indeed have very few components, thus making it easy to say what is important vs not. I would like authors to counter this point and provide examples that they discovered that might not have been that obvious.
- Few things are a bit unclear, as technique 1 has skip connections , but the authors later go to say skip connections do not contribute to the performance improvement. Can authors remove skip connection from technique 1 and show the curves again.
- Further since training algorithm (AdamW) in this case plays a very important rule, it would be helpful to analyse it more closely by checking previous versions of Adam such as RMSprop or even before.
- Same for data augementation, a more indepth study regarding this would be useful.

**Questions:**

Some qs are in the Weakness section.
- Can authors compare on the benchmarks on which Pi-Sigma paper had shown their result? This will help realise if the improvements are indeed benchmark specific or general.
- What is the role of convolutions vs MLP mixer. I think the standard convention is that if u have less data you should do convolution, however this is not clearly indicated from the author's experiments.

---

> ### Author Response · Authors · 2023-11-16
> **Response to Reviewer jSdE (1/2)**
>
> We are grateful to the reviewer jSdE for the attentive study of our work and their appreciation for the scrutinization of different components we conduct. We address their concerns below. The resulting changes in the paper are highlighted with red color for visual distinction. If there are any remaining unclear points, we would be happy to discuss them further.
>
> > **Q1**:  The current architectures of MLP mixer or ViT indeed have very few components.
>
> Let us explain why both MLP-mixer and ViT rely on *a lot of components* and in fact the contribution of such components is not well-studied. Concretely:
>
> * In MLP-Mixer that the reviewer mentions, they use Adam as an optimizer for the pre-training (which is the most important contributor for their final accuracy) and never question the impact of the optimizer. There are a number of additional details that are important. For instance, the pretraining on JFT-300M, which is a proprietary dataset. We do believe that assessing the components in isolation can offer fresh insights. The pre-training implementation details as printed from the paper are [here](https://imgur.com/a/mzYBf4Q).
>
> * In ViT, many of the aforementioned components are still valid. They use Adam for pre-training and JFT-300M for pre-training.
>
> In addition, both the MLP-Mixer and the ViT have plenty of skip connections. Even though the presence of skip connections is prevalent across different architectures, to our knowledge, the role of skip connections is still not conclusively settled as we point out in sec. 3.8.
>
> Therefore, we do believe that the true value of the components can be obfuscated by the use of a lot of implementation details. This is precisely why we believe that exploring the components in an isolated network can provide a more valuable guide for their generalization.
> _____
>
> > **Q2**: Technique 1 has skip connections, but the authors later go on to say skip connections do not contribute to the performance improvement.
>
> We are thankful to the reviewer for the attentive reading. Let us clarify what we meant: In technique 1 we intended to add only the deep Pi-Sigma blocks. However, we noticed that this was not trainable without skip connections. Therefore, we do add the skip connections along with the deep Pi-Sigma.
>
> Inspired by the question of the reviewer, we assess whether we can remove or tone down the skip connection influence in the network. We add a scalar $\rho$ to scale the influence of the skip connections across the network. The $\rho$ is sampled from a Gaussian distribution with a mean $\mu$ and a variance of $10^{-3}$. The mean starts at $1$ in the beginning of the training, which means that we use the regular (unscaled) skip connection and then the mean decreases during the training, therefore decreasing the contribution of the skip connection. Interestingly, the accuracy of the network seems to be increasing up to the point that the mean value becomes $0.04$. This experiment is now described in sec. B.7 along with the corresponding accuracy figure [here](https://imgur.com/a/BieIj4T).  It is possible that a more refined analysis could further increase the performance of the network and/or completely remove the skip connections, however this is not our main goal.
>
> Lastly, in order to avoid any further confusion for the interested reader, we change the claim in the introduction to: “Skip connections seem to offer advantages in terms of optimization, but their role in the generalization should be further scrutinized.”
> _____
>
> > **Q3**:  Analyze more closely the role of the optimizer.
>
> We are thankful to the reviewer for the suggestion. We already included a version using pure SGD in the original submission (typically the first column/row of results in ablation studies such as in Fig. 2 and Tables 7, 13). However, we conduct a dedicated study where all components remain in-tact apart from the optimizer. The results with various versions of Adam, along with SGD and RMSProp are indicated below (when trained on Cifar10):
>
>
> |      	| AdamW | Adam | AdamP | SGD  | Lars | RMSProp |
> |----------|-------|------|-------|------|------|---------|
> | Accuracy | 88.0  | 84.5 | 84.7  | 37.5 | 29.4 | 87.8	|
>
> We have added the related results in sec. B.8. Once again, we are thankful to the reviewer for strengthening our observations.
> _________
>
>
> > **Q4**: Analyze data augmentation more in-depth.
>
> We appreciate the suggestion. We have conducted the following ablation study: We assess the performance when one of the augmentation methods is removed and we obtain the following results:
>
> |      	| No Auto augment | No Mixup | No smoothing | No random erase | Baseline |
> |----------|-----------------|----------|--------------|-----------------|----------|
> | Accuracy | 86.0        	| 84.6 	| 85.4     	| 85.4        	| 88.0 	|
> ______

---

> ### Author Response · Authors · 2023-11-16
> **Response to Reviewer jSdE (2/2)**
>
> > **Q5**: Is it possible to reproduce the original experiments of Pi-Sigma?
>
> To our knowledge the dataset of “underwater SONAR transient data from the DARPA data set” is not available anymore. If the reviewer has any link to that, we would be happy to reproduce the results.
>
> However, beyond the architecture, we also believe that the probabilistic learning rule that the original paper used might insert another complication into this comparison. We would be happy to discuss this further if the reviewer believes this is very important.
> ______
> > **Q6**: Role of convolution vs MLP-mixer in the limited data regime.
>
> We do agree in principle that having less data requires additional inductive bias in order to guide the learning of the network. However, our point here is precisely to emphasize that various components added to the architecture, e.g., layer normalization or data augmentations, alter significantly the inductive bias of the network and result in a substantial improvement in the performance. We do believe this is an understudied topic that warrants a careful analysis from the community.
>
>
> We are thankful for the time and effort of the reviewer. If there are any further questions, we are more than happy to clarify.

---

### Author Response · Authors · 2023-11-19
**New experiment and insights on the confidence of the classifier**

Dear reviewers,

inspired by the request to offer new insights that guide the model design of future components, we have added a new section in the paper. The new section examines the understudied relationship between component design and confidence of the classifier. The confidence of the classifier is a key component and many papers focus on the over-confidence of neural networks, which might be an issue particularly in real-world tasks.

Concretely, the following experiment is conducted and can be found in sec. C in the revised manuscript:

The experiment is designed as follows: We train the networks using the different techniques on Cifar-10 as described in sec.3.2 and utilize the pre-trained networks. We utilize only the datapoints that have a correct prediction and obtain the confidence of the classifier (i.e., the output of the softmax) for each datapoint. Then, we simply plot the distribution as quantized in 10 bins. The results in [Fig.7 of the paper](https://imgur.com/a/SNYu3YY) exhibit the following interesting patterns: (a) layer normalization has a major impact on the confidence of the classifier, while (b) Technique 6 (i.e., transpose block) also has a major impact on the confidence. The impact of the transpose block can be attributed to the sharing of information across tokens through the block.

On top of the already obtained insights, we do not believe we have noticed this insight in previous work. If any of the reviewers is aware of a similar insight, we would be happy to study it and enrich our experiments and improve our paper.

Best,

Authors

---

> ### Author Response · Authors · 2023-11-22
> **Extension of the confidence for classification under noise**
>
> Dear reviewers,
>
> in order to extend further our method and insights we extended the newly added experiment to robustness. Concretely, we add one technique at the time (as we do above) and we consider the Gaussian noise setting from Cifar10-C. The results depict an interesting separation from the aforementioned experiment. Namely, the normalization has a less dramatic change in the confidence and in general the confidence changes when adding different techniques are milder.
>
> Since we are entering the last day of discussions, we would appreciate the feedback of the reviewers.

---

> > ### Author Response · Authors · 2023-11-23
> > **End of discussion**
> >
> > Dear Reviewers and ACs,
> >
> > we appreciate your time in handling our submission. As we are approaching the end of the discussion period (Authors - Reviewers), we are confident that our responses address the major concerns expressed in the original reviews. We hope that this is taken into account, along with the general consensus that there are some novel insights (already in the original paper). During the rebuttal, we have added new experiments and clarifications that have augmented those insights and to our knowledge those new insights have not been referenced before in the literature. Therefore, we would appreciate it if this is reflected in your evaluation scores.
> >
> > Best,
> >
> > Authors

---

### Meta-Review · Area_Chair_VLXS · 2023-12-05

**Metareview:**

The paper investigate an old architecture, Pi-Sigma, and proposes to leverage a few modern tricks to improve the performance so that it would be comparative to modern architectures. While the reviewers acknowledge the paper has some merits in terms of interesting techniques, extensive experiments as well as good presentation. The authors also provide inadequate motivations for Pi-Sigma network, and the resulting architecture does not bear any comparative advantage over existing ones. Furthermore, the authors didn't provide sufficient guiding principles for future architecture design as well. Overall, it is good that an old architecture has made working to a reasonable level. But why should one care? What does that imply? It's authors' job to convey that clearly and convince readers. Unfortunately, I was left with an impression that it is just because it's an 30-year-old architecture, which was not really a good enough justification.

**Justification For Why Not Higher Score:**

According to the reviews and my personal read, it looks like the proposed method may not have a very significant impact on the community, or at least the paper has not done a good job conveying that.

**Justification For Why Not Lower Score:**

n/a

---

### Decision · Program_Chairs · 2024-01-16

Reject